# Global wood fuel production estimates and implications

E. Ashley Steel [1] ✉, Oliver Stoner [1,2], Harry Podschwit [1], Bruno Paz [1,3], Ilaria Bombelli [1,4,12], Sophia L. Simon [1,13], Erin Peterson [1,5], Holger Weimar [6], Sebastian Glasenapp[6], Richard Sikkema [7], Nazik Elhassan [8], Rob Bailis [9], Florian Steierer[10] & Leonardo R. Souza[11]

Global wood fuel production can indicate opportunities and also challenges in sustainable development, forest management, and energy access. Estimates of wood fuel removals and charcoal production are essential for tracking global goals yet reliable measurements are rare. We synthesize existing understanding through a mechanistic, conceptual model and build on it to develop statistical models from official statistics and over 2000 newly identified data points. For 2019, we estimate 2525.7 million m³ of wood fuel removals globally, approximately 30% higher than previously understood. Our estimates are 50% higher in Africa and 40% higher in Asia, 10% lower in the Americas and 20% lower in Europe. Global production of wood charcoal is estimated at 70.5 million tonnes, approximately 50% higher than previous values; our estimates are 20% higher in Africa and 200% higher in Asia. These estimates describe global shifts in wood fuel removals and charcoal production and improve our understanding of the forest sector; they will likely underlie global models used to forecast future trends.

Wood fuel production and consumption are intimately linked with local economies; forest management; human health; renewable energy; and sustainable development. Wood fuel is thought to account for roughly half of global annual wood removals from forests and from trees outside forests. Understanding wood removals for fuel and for wood charcoal production, which can exceed removals for industrial wood products[1] in some countries, is essential for designing strong policies to support sustainable energy production and natural resource management.

Traditional wood fuels, e.g., sticks, branches, chopped logs, and wood charcoal, are essential for those who cannot access or afford other energy sources, yet unregulated or unsustainable production often results in deforestation and forest degradation[2]. Both are linked to climate change and loss of ecosystem services such as healthy soils, natural flow regimes, and biodiversity[3,4]. The combination of pollution from incomplete combustion and carbon lost from unsustainable practices and land degradation results in 1–2 billion tons of greenhouse gas emissions each year, ~ 2% of global emissions[5]. Carbon dioxide emissions from household food preparation with wood fuels harvested in a non-renewable manner were estimated at around 745 million tons in 2019[6]. Forest harvests between 2010 and 2050 are expected to bring net emissions between 3.5 and 4.2 billion tons of carbon dioxide[7]. An estimated 30% of wood fuel harvest and removal in pantropical regions is unsustainable, with

[1]Forestry Division, Food and Agriculture Organization of the United Nations (FAO), Rome, Italy. [2]School of Mathematics and Statistics, University of Glasgow, Glasgow, Scotland, UK. [3]Department of Agricultural Economics and Rural Development, University of Göttingen, Göttingen, Germany. [4]Department of Statistical Sciences, Sapienza University of Rome, Rome, Italy. [5]EP Consulting, Brisbane, QLD, Australia. [6]Thünen Institute of Forestry, Hamburg-Bergedorf, Germany. [7]Tall Forester Trees (advisory services), Heteren, the Netherlands. [8]International Renewable Energy Agency (IRENA), Abu Dhabi, United Arab Emirates. [9]Stockholm Environment Institute, Somerville, MA, USA. [10]United Nations Economic Commission for Europe, Geneva, Switzerland. [11]United Nations Statistics Division, New York, NY, USA. [12]Present address: Italian National Institute of Statistics, Rome, Italy. [13]Present address: GreenCollar, Sydney, NSW, Australia. ✉e-mail: Ashley.Steel@FAO.org

~275 million people living in wood fuel depletion hotspots across South Asia and East Africa[5].

Inefficient use of wood for household energy contributes to household air pollution, the third leading risk factor of global disease burden[8], and exacerbates gender inequality[9] through the tremendous opportunity cost of time-consuming traditional wood fuel collection. Future projections suggest that 23% of the world's population (1.9 billion people) will lack access to clean cooking in 2030, and that the Sub-Saharan African population relying on polluting fuels will exceed 1 billion by 2025[9,10]. Conversely, modern wood-based fuels, including wood pellets and other agglomerates that are tracked separately in FAOSTAT[1] but are interlinked, are typically used for residential, district or industrial heating and electricity generation, as well as the cogeneration of heat and power (CHP)[11]. They are associated with sustainable forest management and climate change mitigation through decreasing dependence on fossil fuels and accelerating progress toward net-zero emissions[12].

Reliable wood fuel production estimates play a key role in tracking national and international progress toward the Sustainable Development Goals (SDGs) and nationally determined contributions (NDC's) defined under the Paris Agreement. Wood fuel production is linked to SDG 7 on affordable, reliable, sustainable and modern energy as well as goals supporting good health, gender equality, climate action, and the protection, restoration and sustainable use of terrestrial ecosystems[13]. Wood fuel estimates are a foundation of indicator 7.2.1—the share of renewable energy in the total final energy consumption—and are vital for tracking the share of wood-based energy in total final energy consumption, Indicator 10 of the Global Core Set of Forest-related Indicators[14], and Indicator 3 of the Global Bioenergy Partnership on harvest levels of wood resources[15].

Estimates of wood fuel and charcoal production serve as inputs for a range of high-level models. In global forest sector models, e.g., Global Biosphere Management Model[16], Global Forest Products Model[17], and the Global Forest Sector Model of the European Forest Institute[18], fuelwood demand is often based on FAOSTAT wood fuel values for historical periods. The Forest Resources Outlook Model, used to estimate global trade in forest products and changes in forest area, also uses FAOSTAT values as inputs[19].

Many countries find it difficult to produce statistics on the production and use of wood fuel[20]. Reliable measures of wood fuel production are expensive due to the need to account for informal markets, including unregistered and sometimes illegal production, and direct household collection of wood fuels. In countries where these challenges are more prominent, often countries with low gross domestic product (GDP) per capita, time- and resource-intensive household surveys are needed to capture unofficial production and consumption of wood fuels. Though the precision of data from household surveys has rarely been quantified, they are likely imprecise given the challenges of data collection across dispersed and difficult to access heterogeneous areas with multiple local languages[21,22]. Consequently, available data are sparse.

The Food and Agriculture Organization of the United Nations (FAO) provides global data on forest product production and trade, including removals of wood for fuel and wood charcoal production, by country (or territory) and year through a publicly available online platform, FAOSTAT. Data submitted by countries or territories to FAO and partners are considered official and the best available information, although they are estimates of varying and unknown quality. Note that when referring generally to countries and territories in this paper, we simply say countries but when referring specifically to data, estimates and maps, we maintain the more formal and correct definition of countries and territories. As of 2019, only 72 of the 217 countries or territories for which FAO provides values on forest product production through FAOSTAT had provided non-repeating and therefore informative official data for at least 10 years since 1960. Where

official data are unavailable, FAO provides extrapolated estimates from a linear regression model developed in 2005.

Meanwhile, complimentary data can be obtained from energy balances, consumption estimates from household surveys, other information collected in previous years or similar countries, or remote sensing of charcoal kilns or forest change[23]. The need for accurate and reliable estimates motivates a desire to take advantage of all available information and apply a modelling framework that can integrate disparate data types, borrowing information across similar years and countries to provide reliable estimates where official data are lacking. We bring information collected and statistical developments from the last 20 years to advance these wood fuel removal and wood charcoal production estimates.

In this paper, we produce country-level annual estimates of wood fuel removals and charcoal production, alongside aggregated regional and global estimates. To achieve this, we bring together existing information through a modelling framework built from our combined understanding of the mechanisms driving wood fuel production. Our contribution combines three pieces of work: first, a systematic country-by-country data search resulting in a dataset of over 2000 data points of wood fuel and wood charcoal production and consumption from 1999 to 2019; second, a conceptual model of the drivers of wood fuel production and consumption across countries, as a foundation for modelling; third, a flexible, non-linear machine-learning approach to predict per capita wood fuel demand and the fraction of that demand met with charcoal.

## Results

### A conceptual model of drivers of wood fuel demand and production

Our conceptual model (Fig. 1) was developed and refined by a multidisciplinary, multi-agency team (see Methods) and provides a guide for model-building. In this framework, the size of the population, the area of various types of land, and the size of the economy (GDP) are the key drivers of household and industry energy demand (consumption) at the national level. Then poverty, availability of alternatives, and availability of wood drive use of wood fuels obtained through national production or trade to meet this demand. Changes in forest area and cover might be correlated with wood fuel removals or charcoal production and were therefore also included in the conceptual model.

Predictor variables used in modelling (Supplementary Table 1) were first identified using the conceptual model. Although not all variables in the model were available at a global scale, the model elucidates the types of information most likely to be useful in future modelling efforts, revisions or for models at finer scales. It also prevents collection of large amounts of unnecessary information and reduces the risk of model overfitting or spurious correlations. Compilation and preparation of covariates used in modelling (see Methods) is described in Supplementary Information.

### Clustering countries and territories based on expected indicators of wood fuel production

We developed clusters of countries and territories (Fig. 2, Supplementary Table 2) using a time series clustering algorithm based a suite of ten variables (Supplementary Table 1) identified from the conceptual model (Fig. 1, see Methods). We assume that countries and territories that are similar with respect to these ten variables also share similar mechanisms driving energy demand and the fraction of that demand met with wood fuel. Creating models by cluster then enables pooling of information between similar countries and territories and stronger predictions for countries and territories with insufficient data. Although several of the ten variables have spatial patterns, variables specifically describing country location, e.g., longitude, latitude, and region, were not included in the clustering algorithm. It is therefore noteworthy that the resulting clusters have clear geographic structure.

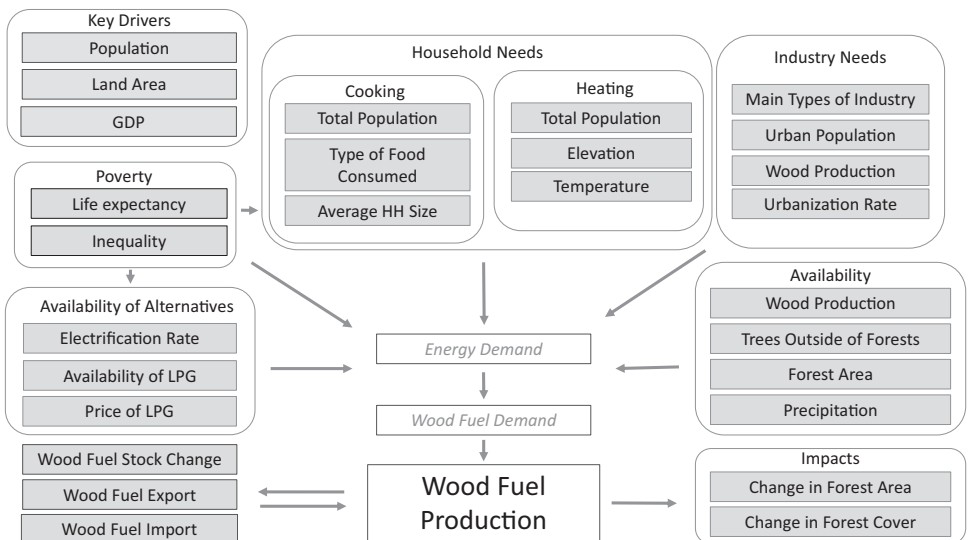

**Fig. 1 | The conceptual model for wood fuel production, describing the mechanistic drivers of wood fuel removals at the national level.** These boxes do not represent the predictor variables in the model but rather the theoretical foundation for selection of predictor variables from globally available data. Unshaded headings describe a first level assessment of drivers or types of indicator variables, e.g., key drivers, household needs, or impacts. Shaded boxes describe specific indicators of these drivers for which availability of global data was assessed. Final selection of predictor variables for clustering and for modelling is described in the Methods. Predictor variables used are referenced in Supplementary Table 1. GDP Gross domestic product; LPG liquid petroleum gas.

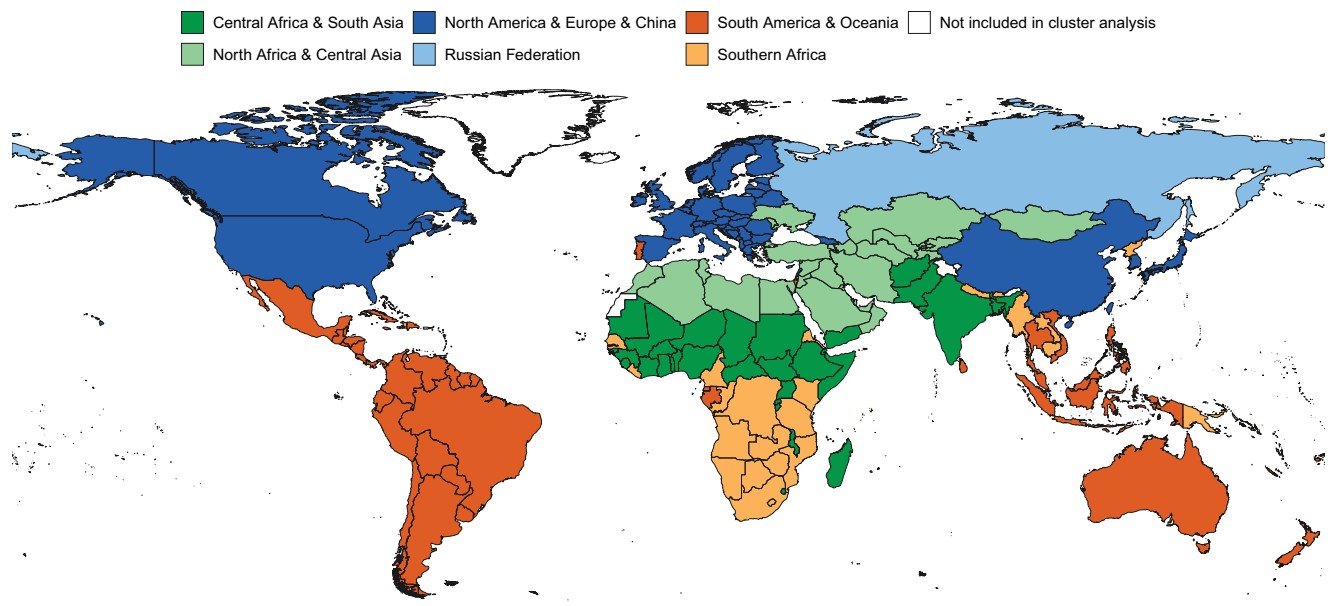

**Fig. 2 | Country and territory clusters used in modelling (Supplementary Table 2).** Note that cluster names are for simplicity only; all countries included in each cluster are listed in Supplementary Table 2. Countries or territories displayed in white were not included in the cluster analysis (see Methods). Country and territory boundaries are based on 2020 United Nations Geospatial mapping data[55]. The boundaries and names shown and the designations used on these maps do not imply the expression of any opinion whatsoever on the part of the Food and Agriculture Organization of the United Nations (FAO) concerning the legal status of any country, territory, city or area or of its authorities, or concerning the delimitation of its frontiers and boundaries. Dashed lines on maps represent approximate border lines for which there may not yet be full agreement; colors chosen for each cluster are for visual clarity only and have no meaning.

## Estimates of wood fuel removals

We assessed out-of-sample predictive accuracy of the wood fuel demand models using 10-fold cross-validation, repeated 10 times (Supplementary Information). Results indicated that models fit relatively well overall for out-of-sample data (Supplementary Fig. 5), with the mean absolute errors ranging from 0.039 m³ per capita in North Africa and Central Asia to 0.201 m³ per capita for Central Africa & South Asia (Supplementary Table 5).

We estimate that 2525.7 million m³ of wood fuel were removed in 2019 (Table 1), comprising 1076.7 million m³ in Africa, 301.0 million m³ in the Americas, 990.5 million m³ in Asia, 140.2 million m³ in Europe and 17.4 million m³ in Oceania (Table 2). Annual national estimates and annual continental summaries are provided in Supplementary Data 1 and 2.

Estimated wood fuel removals were naturally low (<0.05 m³ per capita) in parts of the world with little forest cover, including

**Table 1 | Global total annual estimated wood fuel removals (million cubic meters (m³)) and charcoal production (million tonnes)**

| | Wood fuel (million m³) | | | | Charcoal (million tonnes) | | | |
|---|---|---|---|---|---|---|---|---|
| | Modelled countries | | All countries | | Modelled countries | | All countries | |
| | FAOSTAT | This study | FAOSTAT | This study | FAOSTAT | This study | FAOSTAT | This study |
| 1999 | 1747.6 | 1974.9 | 1815 | 2023.9 | 36.3 | 50.2 | 37.1 | 50.2 |
| 2000 | 1764.3 | 2026.7 | 1795.2 | 2039.2 | 35.9 | 51.2 | 36.7 | 51.3 |
| 2001 | 1771.3 | 1936.5 | 1802.5 | 1949.5 | 37.1 | 50.4 | 37.9 | 50.4 |
| 2002 | 1788.8 | 1947.2 | 1820.6 | 1960.2 | 39.2 | 51.6 | 40.0 | 51.7 |
| 2003 | 1798.9 | 2017.5 | 1831.8 | 2031.3 | 41.5 | 55.5 | 42.4 | 55.6 |
| 2004 | 1802.4 | 2039.4 | 1836.2 | 2053.6 | 42.2 | 57.6 | 43.1 | 57.6 |
| 2005 | 1791.4 | 2034.8 | 1825.3 | 2049.2 | 43.0 | 57.2 | 43.9 | 57.3 |
| 2006 | 1801.8 | 1970.6 | 1834.8 | 1985.7 | 43.8 | 56.4 | 44.8 | 56.5 |
| 2007 | 1806.8 | 1994.4 | 1841.2 | 2010.7 | 45.8 | 58.4 | 46.8 | 58.5 |
| 2008 | 1817.2 | 2020.2 | 1848.4 | 2033.0 | 45.9 | 61.0 | 46.9 | 61.1 |
| 2009 | 1809.1 | 2048.5 | 1840.2 | 2061.1 | 44.4 | 61.5 | 45.5 | 61.5 |
| 2010 | 1831.3 | 2083.3 | 1864.0 | 2097.2 | 45.4 | 62.4 | 46.5 | 62.5 |
| 2011 | 1836.4 | 2138.5 | 1871.0 | 2154.2 | 46.8 | 64.8 | 47.7 | 64.9 |
| 2012 | 1872.8 | 2237.9 | 1887.4 | 2252.5 | 49.1 | 71.0 | 49.2 | 71.1 |
| 2013 | 1883.7 | 2284.7 | 1897.8 | 2298.7 | 50.7 | 74.7 | 50.8 | 74.8 |
| 2014 | 1885.7 | 2331.1 | 1900.4 | 2345.8 | 51.5 | 76.5 | 51.6 | 76.5 |
| 2015 | 1886.3 | 2384.7 | 1901.3 | 2399.7 | 51.1 | 77.0 | 51.2 | 77.1 |
| 2016 | 1912.2 | 2448.9 | 1927.8 | 2464.5 | 50.7 | 77.0 | 50.8 | 77.1 |
| 2017 | 1916.3 | 2493.9 | 1931.1 | 2508.7 | 51.5 | 77.5 | 51.6 | 77.6 |
| 2018 | 1929.6 | 2505.6 | 1946.0 | 2522.0 | 53.1 | 78.5 | 53.2 | 78.5 |
| 2019 | 1925.1 | 2510.5 | 1940.3 | 2525.7 | 53.4 | 79.8 | 53.4 | 79.8 |

Columns compare values currently reported in FAOSTAT and estimates from the random forest models in this study (This study). Global estimates are aggregated separately for only those countries modelled (Modelled countries) and for all countries and territories for which data are reported in FAOSTAT (All countries), including all modelled countries as well as Christmas Island, Cook Islands, Curaçao, Faroe Islands, Gibraltar, Greenland, Iceland, Malta, Marshall Islands, Monaco, Nauru, Russian Federation, San Marino, Tokelau, Tuvalu, and Western Sahara, where data were available.

North Africa, the Middle East, and Central Asia (Fig. 3). Countries and territories with the highest per capita removals were mainly situated in Sub-Saharan Africa and North-East Europe.

These estimates of 2019 wood fuel removals are 585.5 million m³ (30.2%) higher at a global scale (Table 1) than what is currently reported in FAOSTAT. This net increase comes from our estimates being 370.6 million m³ (52.5%) higher for Africa, 37.5 million m³ (11.1%) lower for the Americas, 278.0 million m³ (39.0%) higher for Asia, 32.9 million m³ (19.0%) lower for Europe, and 7.4 million m³ (73.8%) higher for Oceania (Table 2, Fig. 3). Differences between our estimates and current FAOSTAT values are also mixed directions within continents. For instance, our estimates of 2019 wood fuel removals are 87.0 million m³ (130.7%) higher for Nigeria and 52.2 million m³ (46.2%) higher for Ethiopia but 26.2 million m³ (54.9%) lower for Ghana. Similarly, our estimates are 110.9 million m³ (36.7%) higher for India and 99.7 million m³ (62.4%) higher for mainland China but 11.4 million m³ (28.2%) lower for Indonesia (Supplementary Data 1).

Estimated trends in wood fuel removals over time were similar to current FAOSTAT values for the Americas, Europe, and Oceania, but our estimated values show a much faster increase in Africa through the 2010s and a relatively stable pattern for Asia as compared to the consistently decreasing trend in current FAOSTAT values (Fig. 4a). Considering per capita removals for only the rural population did not change the interpretation of results (Fig. 5a).

The most important predictors for modelling wood fuel demand per capita (Fig. 6a, Supplementary Fig. 2) varied by cluster, suggesting differing mechanisms driving wood fuel demand across regions. However, the two climate variables, Minimum Temperature and Rainfall, were important across all clusters. In Southern Africa, variables describing the proportion of the population mainly using biomass fuels (including wood) and charcoal for cooking, Biomass Use and Charcoal Use, were also particularly important. In South America

and Oceania, inequality, as captured by Gini Index, had the highest importance score. In both Central Africa & South Asia and Southern Africa, the two variables indexing the forest sector, other industrial roundwood production (OIR) and Sawlogs, were notable.

## Estimates of charcoal production

Results from the tenfold cross-validation assessment for charcoal models (Supplementary Information) suggested a relatively clear relationship between the fraction of wood fuel demand met by charcoal data and the out-of-sample predictions (Supplementary Fig. 7). Mean absolute errors ranged between 4.24 percentage points in North America and Europe and China to 13.3 percentage points in North Africa and Central Asia (Supplementary Table 5).

We estimate global charcoal production at 79.8 million (metric) tonnes in 2019 (Table 1), of which 40.6 million tonnes were produced in Africa, 10.5 million tonnes in the Americas, 27.1 million tonnes in Asia, 1.1 million tonnes in Europe, and 0.5 million tonnes in Oceania (Table 3, Fig. 7, Supplementary Data 2). Looking at per capita charcoal production, there is a visual divide between the southern and northern hemispheres (Fig. 7). Of the ten countries or territories with the highest per capita charcoal production in 2019, nine were in Africa, and all were estimated to produce more than 50 kg of charcoal per person.

Our global estimate of 2019 charcoal production is 26.4 million tonnes (49.3%) higher than what is reported in FAOSTAT. Our estimate for Asia, for example, ranges from 2.4 to 3.2 times that of what is currently reported in FAOSTAT over time and the difference increases over the last decade (Fig. 4b, Table 3). Furthermore, while our estimates of total charcoal production in Africa are lower than current FAOSTAT values toward the start of the 20-year analysis period, they show a sharper increasing trend and, by 2019, these estimates are 17.8% higher. This increasing trend potentially reflects charcoal's

**Table 2 | Annual estimated wood fuel removals (million cubic meters (m³)) by continent (FAOSTAT continent code included in parantheses)**

|      | Africa (5100) | | Americas (5200) | | Asia (5300) | | Europe (5400) | | Oceania (5500) | |
|------|---------|------------|---------|------------|---------|------------|---------|------------|---------|------------|
|      | FAOSTAT | This study | FAOSTAT | This study | FAOSTAT | This study | FAOSTAT | This study | FAOSTAT | This study |
| 1999 | 536.6   | 573.3      | 311.3   | 314.5      | 813.8   | 979.4      | 140.5   | 140.3      | 12.7    | 16.4       |
| 2000 | 551.3   | 601.1      | 314.4   | 320.5      | 807.8   | 996.3      | 109.1   | 104.3      | 12.7    | 17.0       |
| 2001 | 561.3   | 629.1      | 315.6   | 324.5      | 801.3   | 873.2      | 111.8   | 105.3      | 12.6    | 17.4       |
| 2002 | 566.5   | 649.5      | 314.3   | 325.3      | 815.8   | 861.4      | 112.7   | 106.8      | 11.4    | 17.3       |
| 2003 | 575.3   | 678.0      | 316.9   | 322.8      | 807.8   | 902.6      | 120.0   | 110.6      | 11.8    | 17.3       |
| 2004 | 584.3   | 694.2      | 320.5   | 309.9      | 796.4   | 920.4      | 123.5   | 111.9      | 11.5    | 17.1       |
| 2005 | 599.8   | 699.0      | 299.5   | 282.4      | 791.9   | 938.7      | 122.6   | 112.7      | 11.5    | 16.5       |
| 2006 | 610.6   | 706.4      | 296.8   | 270.2      | 786.1   | 874.5      | 130.3   | 117.9      | 10.9    | 16.6       |
| 2007 | 618.4   | 725.4      | 301.8   | 266.8      | 782.5   | 884.5      | 127.5   | 117.3      | 11.1    | 16.7       |
| 2008 | 627.4   | 740.1      | 300.3   | 262.9      | 778.1   | 901.7      | 131.6   | 111.7      | 10.9    | 16.6       |
| 2009 | 635.2   | 761.0      | 289.4   | 257.5      | 772.5   | 913.3      | 132.5   | 112.8      | 10.7    | 16.5       |
| 2010 | 643.6   | 790.1      | 290.5   | 255.1      | 764.4   | 914.8      | 154.8   | 120.5      | 10.7    | 16.7       |
| 2011 | 652.9   | 828.8      | 298.4   | 256.6      | 756.3   | 924.5      | 152.6   | 127.4      | 10.8    | 16.9       |
| 2012 | 657.3   | 903.3      | 305.9   | 256.8      | 748.3   | 943.0      | 165.3   | 132.6      | 10.6    | 16.9       |
| 2013 | 664.1   | 922.5      | 309.7   | 260.5      | 741.6   | 963.9      | 171.8   | 135.2      | 10.6    | 16.7       |
| 2014 | 671.3   | 946.6      | 313.9   | 265.9      | 740.7   | 978.5      | 163.9   | 138.1      | 10.6    | 16.7       |
| 2015 | 679.5   | 986.7      | 307.2   | 269.8      | 735.2   | 986.9      | 169.4   | 139.6      | 10.0    | 16.7       |
| 2016 | 686.4   | 1031.4     | 331.4   | 283.8      | 730.4   | 991.0      | 169.7   | 141.3      | 9.9     | 16.9       |
| 2017 | 693.1   | 1058.5     | 331.3   | 293.7      | 724.8   | 996.1      | 171.9   | 143.3      | 10.0    | 17.2       |
| 2018 | 700.1   | 1066.5     | 340.6   | 299.3      | 718.5   | 994.2      | 176.9   | 144.8      | 10.0    | 17.2       |
| 2019 | 706.1   | 1076.7     | 338.6   | 301.0      | 712.5   | 990.5      | 173.1   | 140.2      | 10.0    | 17.4       |

Columns compare values currently reported in FAOSTAT (FAOSTAT) and estimates from the random forest models (This study). Estimates are aggregated to include all countries and territories for which data are reported in FAOSTAT under each continent code for modelled countries and, to provide complete continental estimates, also including Christmas Island, Cook Islands, Curaçao, Faroe Islands, Gibraltar, Greenland, Iceland, Malta, Marshall Islands, Monaco, Nauru, Russian Federation, San Marino, Tokelau, Tuvalu, and Western Sahara, for the appropriate continent and where data were available.

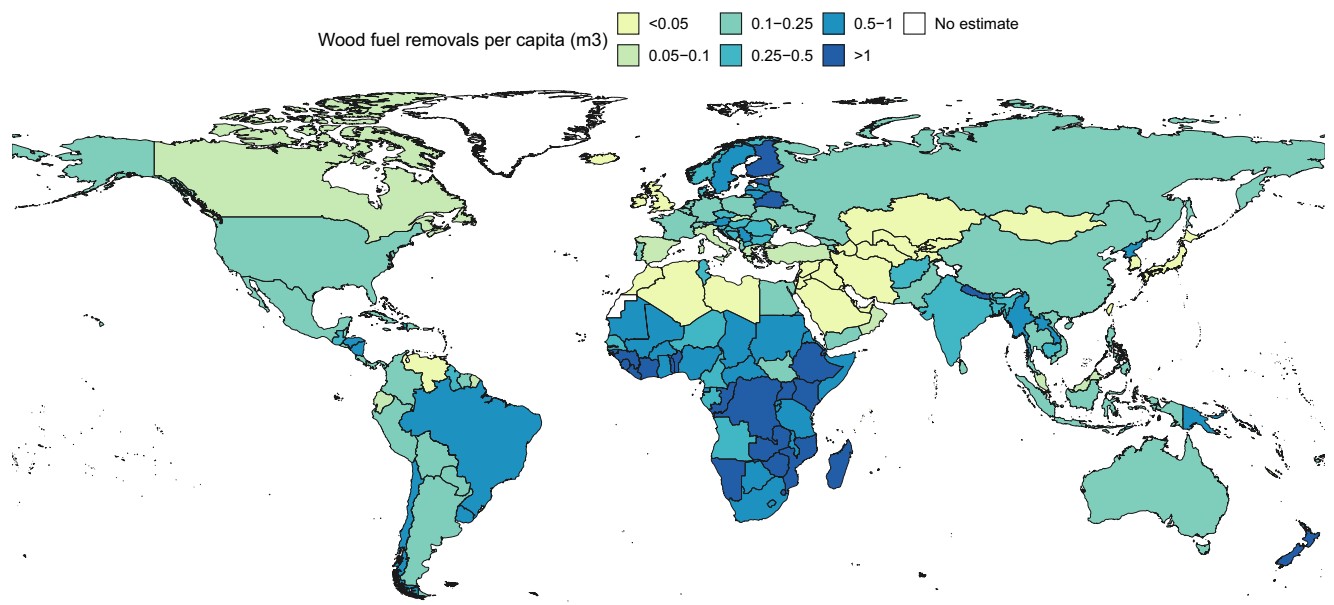

**Fig. 3 | Estimated 2019 wood fuel removals per capita (m³).** We did not produce modelled estimates for Christmas Island, Cook Islands, Curaçao, Faroe Islands, Gibraltar, Greenland, Iceland, Malta, Marshall Islands, Monaco, Nauru, San Marino, Tokelau, Tuvalu, and Western Sahara. Instead, we used official data, where available, or assumed zero wood fuel removals. For the Russian Federation, we used current values from FAOSTAT (accessed 1 November 2024). Country and territory boundaries are based on 2020 United Nations Geospatial mapping data[55]. The boundaries and names shown and the designations used on these maps do not imply the expression of any opinion whatsoever on the part of FAO concerning the legal status of any country, territory, city or area or of its authorities, or concerning the delimitation of its frontiers and boundaries. Dashed lines on maps represent approximate border lines for which there may not yet be full agreement.

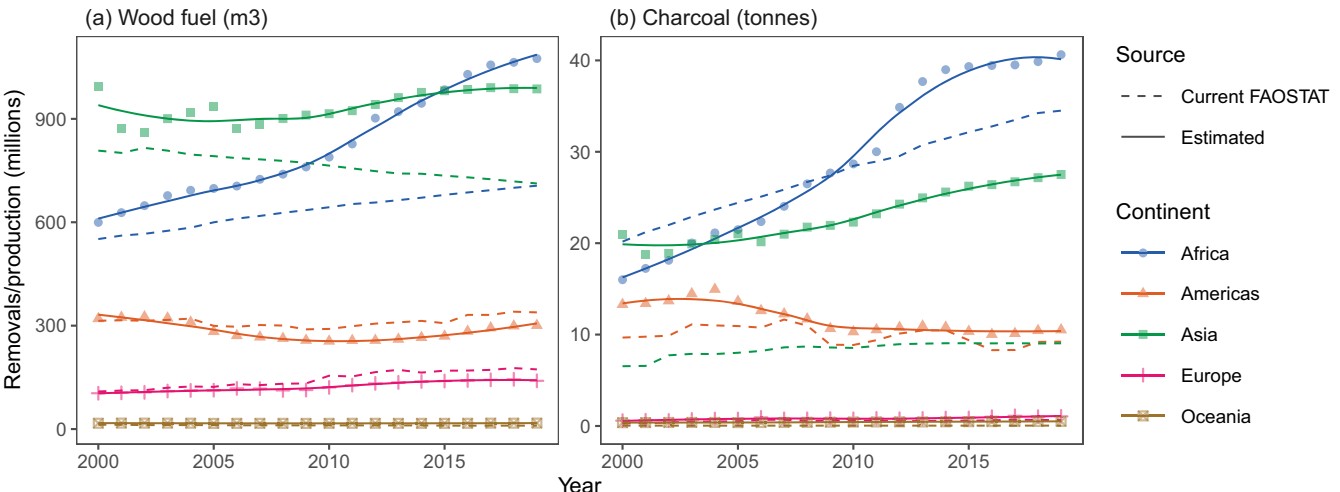

**Fig. 4 | Trends in wood fuel removals and charcoal production.** Estimated wood fuel removals (**a**) and charcoal production (**b**) by continent. Countries included in each continental summary are those in FAOSTAT. Points (shapes) show our estimates and dashed lines show current FAOSTAT values. Solid lines are smoothed long-term trends in the estimates (locally estimated scatterplot smoothing).

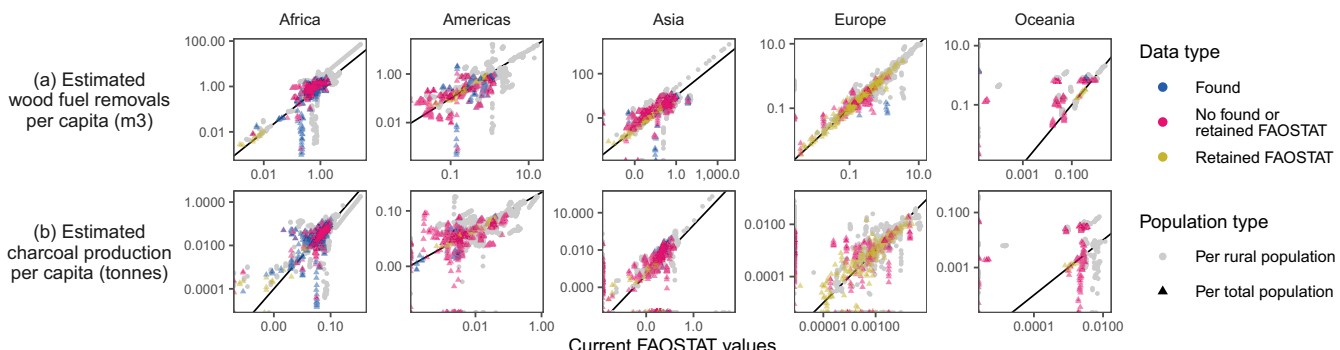

**Fig. 5 | Comparison of current FAOSTAT values and our estimates.** Current FAOSTAT values and our estimates are of **a** wood fuel removals and **b** charcoal production per total capita (triangles) and per rural capita (grey circles). Each point represents one year and country or territory for which estimates were produced. Triangle colors indicate whether values retained from FAOSTAT or found data were available, or whether neither retained nor found data were available on wood fuel demand for each point (year and country or territory).

expanding role as a primary cooking fuel in Africa, particularly in rapidly growing urban areas[10,24]. Comparing our per capita estimates to current FAOSTAT values reveals large differences in Africa and Asia that could not be explained by considering only use in rural areas (Fig. 5b).

The fraction of the population mainly using charcoal for cooking (Charcoal Use) was the most important variable for modelling the fraction of wood fuel demand met with wood charcoal in the Central Africa & South Asia and South America & Oceania clusters (Fig. 6b), which are broadly reflective of the parts of the world where charcoal production per capita is highest (Fig. 7). Production of OIR, an indicator of the forest product sector, also had high permutation importance scores in these two clusters. Minimum Temperature was important across all clusters (Fig. 6b and Supplementary Fig. 3).

## Discussion

Globally, we estimate that ~30% more wood volume is removed from forests, including trees outside of forests, compared to previous FAO values; the distribution of those differences across continents, with over 80% of removals in Africa and Asia in 2019, reflects much higher new estimates of removals in Africa and Asia, and slightly lower new estimates for the Americas and Europe. Meanwhile, we estimate worldwide wood charcoal production at 79.8 million tonnes, about

50% higher than current FAOSTAT values; these estimates are about 20% higher in Africa and 200% higher in Asia.

The full scale of uncertainty in both previous and revised estimates cannot be estimated appropriately from currently available information but is likely high (see Methods). Future work to understand and reduce uncertainties, particularly in the underlying empirical estimates of wood fuel use from household surveys and from energy budgets, will improve our ability to use these values for policy-making and could potentially inform intensified data collection efforts in countries where estimates are most uncertain.

Climate variables, such as Minimum Temperature and Rainfall, were often predominant drivers of both wood energy demand per capita and the fraction of that demand met with wood charcoal. Areas with relatively high levels of rainfall may be more suitable for tree growth, which could be acting as a surrogate for overall wood fuel availability and therefore the possibility of using wood energy. Meanwhile, areas with low minimum temperatures may have a higher demand for wood energy for heating. The importance of these variables suggests a potential for climate change to shift patterns of access and demand.

The proportion of the population predominantly using biomass or charcoal for cooking were also key predictors in both models and across country clusters. Use of these variables connects our modelling

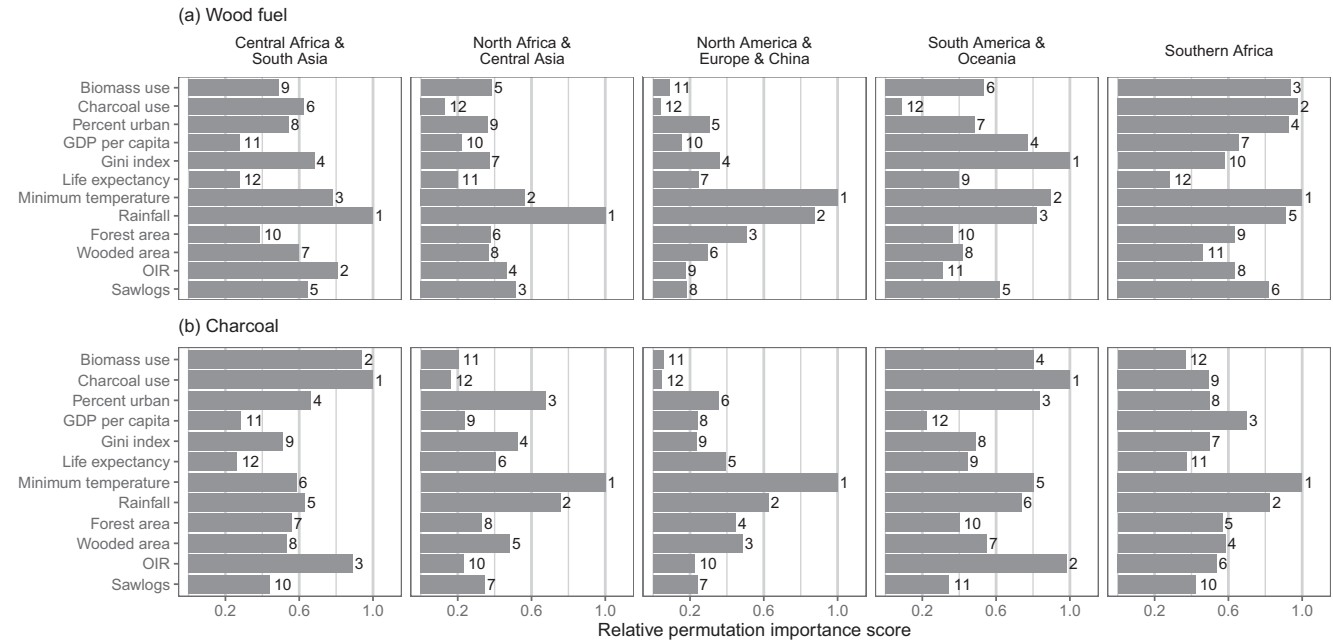

**Fig. 6 | Variable permutation importance scores.** Variable permutation importance scores for the wood fuel demand per capita models (**a**) and fraction of demand met with charcoal models (**b**). Within each cluster of countries, scores are relative to the highest importance score, calculated by dividing each importance value by the highest importance score. Numbers to the right of the bars are the importance score rankings of variables for each cluster.

**Table 3 | Annual estimated charcoal production (million tonnes) by continent (FAOSTAT continent code included in parantheses)**

|  | Africa (5100) | | Americas (5200) | | Asia (5300) | | Europe (5400) | | Oceania (5500) | |
|---|---|---|---|---|---|---|---|---|---|---|
|  | FAOSTAT | This study | FAOSTAT | This study | FAOSTAT | This study | FAOSTAT | This study | FAOSTAT | This study |
| 1999 | 19.87 | 15.31 | 10.27 | 12.86 | 6.56 | 21.09 | 0.38 | 0.61 | 0.02 | 0.34 |
| 2000 | 20.17 | 16.00 | 9.67 | 13.26 | 6.54 | 21.07 | 0.30 | 0.59 | 0.04 | 0.36 |
| 2001 | 21.21 | 17.23 | 9.77 | 13.35 | 6.57 | 18.83 | 0.30 | 0.63 | 0.04 | 0.37 |
| 2002 | 21.99 | 18.14 | 9.87 | 13.65 | 7.73 | 18.93 | 0.38 | 0.60 | 0.04 | 0.38 |
| 2003 | 22.88 | 20.00 | 11.11 | 14.49 | 7.89 | 19.97 | 0.46 | 0.72 | 0.04 | 0.39 |
| 2004 | 23.68 | 21.14 | 11.00 | 14.95 | 7.87 | 20.40 | 0.50 | 0.75 | 0.03 | 0.38 |
| 2005 | 24.42 | 21.49 | 10.93 | 13.62 | 8.01 | 21.02 | 0.51 | 0.77 | 0.03 | 0.37 |
| 2006 | 25.09 | 22.40 | 10.76 | 12.57 | 8.21 | 20.22 | 0.68 | 0.93 | 0.03 | 0.38 |
| 2007 | 25.87 | 24.05 | 11.65 | 12.25 | 8.60 | 21.02 | 0.68 | 0.78 | 0.04 | 0.39 |
| 2008 | 26.70 | 26.52 | 10.97 | 11.71 | 8.69 | 21.71 | 0.53 | 0.78 | 0.04 | 0.41 |
| 2009 | 27.41 | 27.71 | 8.90 | 10.63 | 8.59 | 21.98 | 0.59 | 0.81 | 0.04 | 0.42 |
| 2010 | 28.46 | 28.73 | 8.86 | 10.29 | 8.54 | 22.26 | 0.57 | 0.78 | 0.04 | 0.44 |
| 2011 | 28.93 | 30.04 | 9.39 | 10.49 | 8.76 | 23.05 | 0.55 | 0.83 | 0.04 | 0.46 |
| 2012 | 29.54 | 34.88 | 10.11 | 10.74 | 8.95 | 24.19 | 0.53 | 0.80 | 0.04 | 0.47 |
| 2013 | 30.76 | 37.73 | 10.45 | 10.93 | 9.01 | 24.82 | 0.54 | 0.83 | 0.04 | 0.46 |
| 2014 | 31.44 | 39.03 | 10.48 | 10.82 | 9.05 | 25.33 | 0.58 | 0.88 | 0.04 | 0.47 |
| 2015 | 32.15 | 39.40 | 9.38 | 10.36 | 9.05 | 25.95 | 0.58 | 0.89 | 0.04 | 0.49 |
| 2016 | 32.77 | 39.54 | 8.29 | 10.02 | 9.04 | 26.02 | 0.61 | 1.04 | 0.04 | 0.49 |
| 2017 | 33.49 | 39.56 | 8.32 | 10.13 | 9.05 | 26.23 | 0.70 | 1.14 | 0.04 | 0.50 |
| 2018 | 34.24 | 39.91 | 9.20 | 10.44 | 9.02 | 26.66 | 0.66 | 1.03 | 0.04 | 0.51 |
| 2019 | 34.50 | 40.64 | 9.22 | 10.54 | 9.04 | 27.05 | 0.64 | 1.08 | 0.04 | 0.52 |

Columns compare values currently reported in FAOSTAT (FAOSTAT) and estimates from the random forest models (This study). Estimates are aggregated to include all countries and territories for which data are reported in FAOSTAT under each continent code for modelled countries and, to provide complete continental estimates, also including Christmas Island, Cook Islands, Curaçao, Faroe Islands, Gibraltar, Greenland, Iceland, Malta, Marshall Islands, Monaco, Nauru, Russian Federation, San Marino, Tokelau, Tuvalu, and Western Sahara, for the appropriate continent and where data were available.

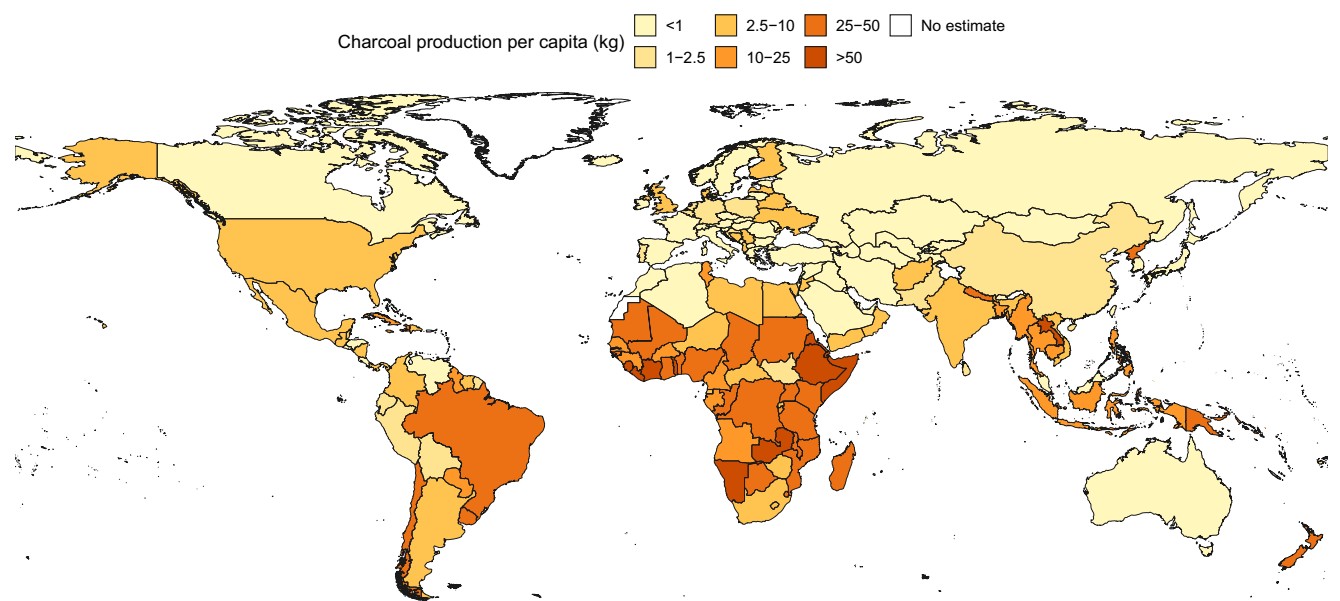

**Fig. 7 | Estimated charcoal production per capita (kg).** We did not produce modelled estimates for Christmas Island, Cook Islands, Curaçao, Faroe Islands, Gibraltar, Greenland, Iceland, Malta, Marshall Islands, Monaco, Nauru, San Marino, Tokelau, Tuvalu, and Western Sahara. Instead, we used official data, where available, or assumed zero wood fuel removals. For the Russian Federation, we used current values from FAOSTAT (accessed 1 November 2024). Country and territory boundaries are based on 2020 United Nations Geospatial mapping data[55]. The boundaries and names shown and the designations used on these maps do not imply the expression of any opinion whatsoever on the part of FAO concerning the legal status of any country, territory, city or area or of its authorities, or concerning the delimitation of its frontiers and boundaries. Dashed lines on maps represent approximate border lines for which there may not yet be full agreement.

to empirical information from a wealth of household survey data on reliance on wood for cooking, which also captures information about a wide range of socioeconomic conditions such as poverty and the availability and affordability of alternative energy sources.

The breadth of predictors important for modelling reflects the complex mechanisms of supply and demand, even where use is primarily based on direct household collection, and agrees with our previous understanding that production and consumption are influenced by the area available for gathering, distance to forest resources, topography, and climate[25]. Several works[26–28] also use country and territory data to predict global fuelwood demand.

Revising estimates of wood fuel removals and charcoal production has implications for tracking sustainable development as they support our estimation of global trends and underlie indicators for monitoring global sustainable development and the emerging bioeconomy, including understanding issues of sustainable supply and demand. National estimates are a foundation of global energy statistics, underpinning our collective understanding of how global energy demands are met, equality in energy access, and progress toward multi-national goals and commitments, including the SDGs[13] and carbon emission reduction goals.

Estimates of wood removals for fuel also support related global emissions accounting. For example, The Global Bioenergy Partnership Indicator 3 on harvest levels of wood resources may be adjusted based on our revised estimates[15]. Such knowledge is key to prioritizing degraded forests for restoration, design of projects to enable efficiencies in the use of wood fuel for cooking, establishment of tree plantations, or incentivising improved utilisation of residues from wood harvesting and processing[29]. Our revised estimates can empower the development of national wood fuel strategies that coordinate interventions and policies across sectors. Malawi's National Charcoal Strategy (2017–2027), for example, provides a framework to address the linked problems of increased deforestation and increased demand for household cooking fuel, while also supporting climate change adaptation and resilience[30].

Increases in estimated wood energy production may be driven by population growth, reductions in household sizes, or newly captured information on household collection of wood and illegal or undocumented production of charcoal. Increases in wood energy production may also be driven by other types of shifts in demand, including increased demand for modern wood fuels, innovations enabling new products or efficiencies in creating existing products such as wood pellets, and altered patterns of global trade. Where estimates of wood energy production have decreased over time, it is likely due to improved access to alternative fuels and urbanization. Differences compared to previous models, which were highly driven by population forecasts, may also reflect differences between forecasted and observed human population trends.

The lack of surveys and measurements in countries where wood energy is most prevalent necessitates the use of modelled estimates for large parts of the world. Where countries do not collect national fuelwood consumption data, international organizations, and non-governmental organizations can provide support to conduct empirical surveys[20,31]. Revisions should naturally occur as new studies and models are developed. Incorporating new empirical data can trigger large revisions, influencing global indicators. For example, a substantial revision between the 2017 and 2018 versions of the SDG 7 tracker[32,33] was based on new empirical data. Revisions can also affect future projections, which have clear implications for policy decisions: for one scenario in the World Energy Outlook, the predicted population without access to clean cooking in 2030 was revised downward from 2.1 billion[34] in the 2021 edition to 1.9 billion people[35] in the 2022 edition.

Our results have implications for global energy and forestry policy. They shift our understanding of how much wood is likely being removed from forests for fuel, of the global distribution of charcoal production, and, potentially, of the distribution of forest labour. Our estimates may yield information about which countries have the most opportunity for wood energy policies to support climate change mitigation. These estimates will also be used as inputs to a variety of

models and analyses that answer questions related to the effect of wood fuel production on deforestation and degradation; the effect of wars and pandemics on wood removals and charcoal production; and the availability of sustainable wood streams to meet forecasted future demand.

Great regional disparities are shown, with wood fuel being particularly critical across Africa where it is responsible for meeting around 40% of total final energy consumption[36,37]. Wood fuel removals have increased over time, particularly in Africa, where they are estimated to have grown by 503.4 million m³ between 1999 and 2019, as compared to a net 1.5 million m³ decrease in the rest of the world. Meanwhile, charcoal production increased by about 25.3 million tonnes in Africa, compared to the net increase of 4.3 million tonnes in the rest of the world, tracking strongly with increasing use of charcoal for cooking in Africa[10].

Our estimates agree with our existing understanding of wood removals for fuel and wood charcoal production across Europe. Despite being slightly lower than current FAOSTAT values, our estimates continue to demonstrate a slightly increasing trend. The trend is likely driven by low-carbon mandates leading to increased demand for pellets[38]. Based on this increasing demand, there has been a shift in the feedstock for producing wood pellets. Where wood pellets were originally produced almost entirely from industrial residues such as sawdust, shavings, and particles, they may now be produced from an assortment of sources including low-quality roundwood[39]. As in the developing world, dependency on wood for fuel also reflects the prices of alternative energy sources; therefore, increased cost of fossil fuels likely contributes to the increasing trend in removals of wood for fuel in Europe[40].

Finally, this work illuminates needs for future work estimating wood fuel removals and charcoal production. The data on which these revised estimates were built pre-date the COVID-19 pandemic, which pushed an estimated 124 million more people into extreme poverty[29] and disrupted global supply chains. Early evidence on pandemic impacts describes shifts in patterns and quantities of wood fuel production through case studies; for example, households in Kenya pivoted from liquid petroleum gas to wood or kerosene for cooking during a pandemic-related lockdown[41]. A modelling approach that could incorporate global disruptions more quickly, e.g., through partial automation, would enable estimates that track global patterns more closely. Building on our approach to revise estimates that pre-date 1999 would be challenging due to a lack of predictor variable data but could also provide a consistent long-term history.

Our model creates an estimate of total wood fuel production, including trees that are harvested specifically for wood energy and branches or deadwood collected both in forests and from trees outside of forests. Separating these sources in future modelling would provide a clearer understanding of wood fuel production sustainability. Capacity building in data collection and continued coherence and collaboration across agencies collecting or working with forest product and energy statistics will also provide continued improvements in our global estimates and understanding of global trends.

It is important to state that the estimates presented here should not be used by countries or territories to revise official submissions to FAOSTAT. Use of these estimates to revise official submissions risks amplifying errors and biases over time. Instead, the estimates can be seen as our best possible prediction where official data based on in-country modelling or in-country empirical data are not available (see Methods).

Energy statistics bear high uncertainty and there are discrepancies between global databases on wood energy from the FAO, the International Energy Agency (IEA), and the UN Statistics Division (UNSD)[42]. A disconnect is also noted between estimates of total biomass consumption in households (assumed mostly for cooking) from IEA and UNSD estimates of biomass fuel use for cooking from the World Health Organization (WHO). Our estimates incorporate modern WHO estimates of reliance on various forms of energy for cooking[10] as predictors, potentially reducing the disconnect between agency models. In the future, including similar estimates of wood fuel use for heating, linking information across agencies through one integrated modelling framework, and quantifying uncertainty in modelled estimates could all further illuminate global patterns and allow more transparent data streams.

## Methods

### Definitions and a brief history of global wood fuel and charcoal modelling

Throughout this paper and analyses, we use the definitions of wood fuel removals and wood charcoal production as found in the FAO Yearbook of Forest Products[43] and, therefore, used in FAOSTAT. Wood fuel removals are defined as "roundwood that will be used as fuel for purposes such as cooking, heating, or power production. It includes wood harvested from main stems, branches, and other parts of trees (where these are harvested for fuel), round or split, and wood that will be used for the production of charcoal (e.g., in pit kilns and portable ovens), wood pellets and other agglomerates." Roundwood includes wood felled or otherwise harvested and removed from forests and from trees outside the forest.

In this paper, we estimate wood fuel production through the demand for wood energy (Fig. 1), and we additionally model the fraction of that demand met by wood charcoal (which we call the "charcoal fraction"). The demand for wood energy is assumed to equal consumption and this is converted to production by adding exports and subtracting imports. Wood charcoal is defined as "wood carbonised by partial combustion or the application of heat from external sources." While it includes charcoal made from shells or nuts, which might add a small source of error to our approach, it excludes bamboo charcoal.

We base our methods on a long history of modelling wood fuel removals and wood charcoal production, which has included an ongoing process of revision and improvement. In 1980, a literature review and search for credible data through country offices identified 265 records of wood fuel consumption for 66 mostly low-income countries or territories[44]. The report of these data, prepared for the UN Conference on New and Renewable Sources of Energy (Nairobi, 1981), noted that reliability of wood fuel production estimates was low, that there was high variability within and between countries or territories, and that there had been no detectable trends in consumption over the previous decade. National consumption was therefore estimated as the average of the values identified, weighted by the size of the population surveyed. For most countries or territories, these average levels of consumption differed greatly from the estimates being used in the FAO Yearbook of Forest Products (e.g., from over 54 times higher in the case of Senegal to only 1 percent of the existing estimate in the case of Mauritania). The data series in FAOSTAT of fuelwood and charcoal production statistics dates back to 1961[1]. Following the literature review in 1980, values were revised backward based on these their revised estimates of per capita consumption, which ranged from <0.001 to 1.12 m³, with charcoal production in tonnes converted as 6 m³ roundwood needed per tonne charcoal produced[45]. Though there were many strengths to this approach, it was based on the average of a limited number of observations per country or territory and fairly simple statistical tools. These per capita estimates provide an interesting reference point but are now obsolete.

Early work estimating and predicting wood fuel consumption was generally based on assumptions about wood fuel that have proved more or less inaccurate over time[46,47]. These assumptions included: (i) overall use consistently increases in proportion to population in developing countries; (ii) production is mostly derived from cutting down trees in forests and is one of the main causes of deforestation and forest degradation; (iii) wood fuel is mostly used for cooking by

poor families in rural areas; (iv) wood fuel use will almost disappear with economic development; and, (v) mechanisms driving production are similar across regions within a country and across countries.

On the demand side, wood fuel consumption has not increased in direct proportion to population in most countries. Instead, structural changes (e.g., urbanization, electrification, and economic development) have led to shifts in consumption patterns. On the supply side, wood fuel production leads more often to forest degradation than deforestation[48], which likely leads to increased supply costs and shifts in consumption patterns or to use of alternate fuels. An important omission in early wood fuel analyses was the importance of trees outside forests for wood fuel supply[47].

In 2005, new estimates and forecasts of wood fuel and wood charcoal production were produced from a study by J. Broadhead, J. Bahdon, and A. Whiteman. The study evaluated past modelling work at FAO and conducted an in-depth literature review of other wood fuel modelling research and available estimates. A set of linear models were used to estimate household wood fuel consumption, non-household wood fuel consumption, and charcoal production. Where adequate official data were available (≥10 years of non-repeating official data), country- or territory-specific models were built using only official data (N = 49 countries or territories for wood fuel; N = 14 countries or territories for wood charcoal). Where countries or territories did not have 10 years of non-repeating official data, models were based on a combination of official data and external data identified during their literature review; depending on how many external data records were identified, a country- or territory-specific model or a model for a group of countries and/or territories may have been produced for a total of 111 linear models. Predictors included GDP, GDP per capita, total population, degree of urbanization, oil production in 1997, forest cover, land area, and average annual temperature; the model of non-household wood fuel consumption also included household wood fuel consumption. Many predictors were natural log transformed.

FAO revised the complete series of wood fuel production figures back to 1961 based on these models. For some countries or territories, the revised estimates varied greatly from those that were previously produced. Forecasts of national wood fuel and wood charcoal production were made until 2030 using forecasted GDP and human population, although details of the exact source and date of these input variables are unavailable. As of 2025, forecasted estimates from the 2005 model revision may be used to produce estimates for countries or territories that do not submit official figures.

## Input data on wood fuel production from FAO
FAO and its partner agencies, the International Tropical Timber Organization, the European Commission's Statistical Office (Eurostat), and the United Nations Economic Commission for Europe (UNECE), annually solicit data on wood fuel removals (including wood for charcoal) (m³) and wood charcoal production (tonnes) from 207 countries and territories via the Joint Forest Sector Questionnaire (JFSQ) and publish them on FAOSTAT. Here, wood fuel production is the aggregate of coniferous and non-coniferous wood fuel production. Countries and territories may, optionally, provide details on the source, accuracy, or precision of their national estimates but such detail is rare. FAO and partners validate reported figures through comparisons with previous years and similar countries or territories, material balance checks, and checking that apparent consumption is not negative. Where there are concerns, FAO or its partner agencies contact the country to identify a resolution. Where countries or territories do not provide official data, FAO and its partners may search published estimates from, for example government websites or non-governmental organization (NGO) project reports. If no published estimates are identified or available, forecasted values from the set of linear models described above are used.

As inputs for our model, we used only non-repeating data from 1999 to 2019 reported by countries and territories or identified in other published sources. At the time of analysis, these values could be identified in FAOSTAT using flags indicating official (reported by countries or territories) or unofficial (other published sources) values. We considered repeated values identical to the first digit (1 m³) to be estimates rather than new information regardless of flag and so we did not use them in model fitting. For wood fuel, it was possible for the flags for coniferous and non-coniferous wood to differ. If the value for either coniferous or non-coniferous wood fuel was ≥60% of the total, we applied that flag to the total. If the proportions of coniferous and non-coniferous wood fuel ranged between 40% and 60%, we applied the least certain flag (official > unofficial > all else). Only these FAOSTAT data submitted by countries and territories or identified from published sources, which we call retained FAOSTAT data, were used in model-fitting to avoid building a new model using estimates from a previous model.

We identified a set of countries and territories (Christmas Island, Cook Islands, Curaçao, Faroe Islands, Gibraltar, Greenland, Iceland, Malta, Marshall Islands, Monaco, Nauru, San Marino, Tokelau, Tuvalu, and Western Sahara) which are reported in FAOSTAT but which have, effectively, zero wood fuel or wood charcoal production. We did not include these countries and territories in modelling. Instead, we simply predicted zero production of either wood fuel or wood charcoal where official data were not available.

## External wood fuel and charcoal data search
For countries or territories without 10 years of non-repeating official wood fuel data, we conducted a systematic country-by-country search identifying over 2000 additional data points published since 1999 on wood fuel or wood charcoal production or consumption. For each country, the search included published literature; governmental websites; and reports from relevant international organizations, national development agencies and NGOs (Supplementary Table 3). For each data point identified, information on data source, data type (e.g., wood fuel, wood charcoal, firewood), unit (e.g., tonnes, m³), level (e.g., per person, per household), sector (e.g., urban/rural, particular industry), spatial scale (e.g., town, region, nation), and temporal scale (e.g., daily, weekly, yearly) were recorded. Another set of over 1000 data points for African countries and territories were collected from the African Energy Commission (AFREC), which provided statistics on wood fuel production from 2000 to 2017 for most of their member states. Hereafter, we refer to all data found during the systematic data search as found data.

The vast majority of found data points were annual estimates at the national level for the total population. Data found for other spatial, temporal, or user scales were transformed where possible. A small number of data points (N < 50) found at a sub-national scale were multiplied by the ratio of the total population for that country or territory and year divided by the total population of the study area (as reported in the data source). All data points found in temporal scales of days or weeks were converted to an annual scale. Additionally, data at the per-user scale were transformed to the national level by multiplying by the total population for that year and country using population data in FAOSTAT. There were a handful of data values at the household level. In these cases, we divided by average household size for the study area as reported by the data source. Finally, we transformed all data into common units of m³ (wood fuel) and metric tonnes (wood charcoal) (Supplementary Table 4).

Before modelling, we eliminated about 30% of the data identified in the systematic literature search. The primary reason data points were eliminated is because they could not be scaled to a national level, e.g., values were apparently reported from permitted sales only, reported on a per-person basis for one city, or results were reported for all bakeries in a country. Data points initially expressed as wood

fuel consumption at the national level were prioritized, as were data points that included all users of wood energy over those that only captured household consumption. If multiple useable data points from the same source were recorded in the found data, the mean was used. Repeated values, where we identified the exact same value year after year for a particular country or territory, were not used because we understood that little new information was added over time. In these cases, only the first three such years were retained as a compromise solution to include what may have originally been a clear understanding of no substantial change. Additionally, found data points resulting in per capita wood fuel consumption estimates that exceeded 10 m$^3$ per year or per capita wood charcoal consumption that exceeded 0.1 tonnes per year were eliminated based on expert opinion. These exceeded reasonable values, reasonable values based on available supply, or evidence from historical estimates, which had a maximum value of 1.12 m$^3$ for wood fuel. Similarly, data points that resulted in near zero per capita consumption for countries or territories where more than 50% of the population is estimated to rely primarily on wood for cooking[10] were eliminated.

About 500 found data points specifically described production or consumption of "firewood", i.e., wood used directly and therefore exclusive of wood harvested for charcoal production. Where possible ($N > 300$), found data on charcoal production/consumption in the same country or territory were converted to the volume of wood and added to the firewood values; this total matches wood fuel removals (as per our definition) more appropriately. About 70% of these firewood values were matched to charcoal production/consumption data from the same source; the remainder were matched to the mean of available charcoal production/consumption data from the same year or the mean from that country or territory over a time window of up to ± 5 years.

Estimates for Nigeria and India were further revised during final model evaluation. In the review of modelled estimates for Africa, we found high volatility in both estimates and input data for Nigeria. Given the large population of this country, we re-opened the data search, identifying additional found data points for wood fuel and wood charcoal. India, a key component of regional (Asia) and global estimates due to its population size, had very few national values for wood fuel or wood charcoal; therefore, we opted to use found estimates for rural and urban populations, reassembling them via annual rural and total population estimates from the World Bank (Supplementary Table 1).

Found data, transformed and scaled for use in modelling are available in Supplementary Data 3. We assumed that all found data represented wood fuel consumption when merging the values from FAOSTAT with found data.

## Expert group and development of the conceptual model

In late 2020, an expert group on wood fuel and modelling was established, which initially consisted of members from UNECE, the International Renewable Energy Agency (IRENA), IEA, FAO's Forestry and Statistics Divisions, La Sapienza Statistics Department, McGill University Statistics Department, UNSD, Ghana Statistical Service, University of Oregon Forestry Department, and WHO. The group informally expanded to include AFREC, University of Glasgow School of Mathematics and Statistics, Thünen Institute of Forestry, and Stockholm Environment Institute.

The expert group met online four times. The first meeting, 30 September 2020, focused on introductions, problem formulation, and initial development of a conceptual model for the drivers of wood fuel and wood charcoal production. The purpose of the conceptual model was to set out what the group felt were the mechanistic drivers of wood fuel and wood charcoal production as a framework for identifying and selecting predictor variables and determining model form. We adopted this approach recognizing that our main goal was to

estimate wood fuel production where there are limited data under current and, where possible, future conditions. We wanted, therefore, to minimize spurious correlations and include only predictors that were physically plausible and explainable.

The second meeting, 29 October 2020, focused on a review of the input data from FAO and of the input data from external sources, as well as revisions to the conceptual model. The third meeting, 10 December 2020, reviewed work on formulating clusters of similar countries and territories (based on the set of variables in the conceptual model), such that countries and territories within each cluster would share a common predictive model. At the fourth meeting, 21 July 2021, the conceptual model was finalized (Fig. 1), a modelling approach was proposed, and predictors were reviewed and either rejected or retained. Price of wood fuel or charcoal, for example, were considered, but it was decided early on not to include price for several reasons. Most importantly, while global price data could be estimated, it would need to be estimated from a model that uses wood fuel and charcoal values from FAOSTAT as the foundation, introducing an invalid circular statistical structure that magnifies biases and errors. While unit prices could be derived in some areas using data from local studies, availability of such empirical estimates are scarce at a global level.

Members of the expert group were consulted informally over the following year and a half to finalize the modelling approach and review the results. An additional meeting of this group is expected after publication to determine the best way forward in revising the values that are publicly available through FAOSTAT.

## Clustering countries and territories to share a common model

To identify suites of countries and territories likely to share similar mechanisms driving wood fuel production and consumption, we applied a clustering algorithm for temporally correlated data to ten predictor variables (Supplementary Table 1) from 199 countries and territories. We used only ten of the possible predictor variables because we first eliminated those that were highly correlated or that represented similar concepts to avoid overweighting the clustering by one idea or mechanism; we selected only one metric to describe rural versus urban population and we selected only Tree Area, which is the sum of Wooded Area and Forest Area and is highly related to country or territory area. We also chose not to use metrics that were directly related to wood and wood fuel production for clustering, so that the clusters could describe countries and territories that shared the same mechanisms driving wood fuel production and not those which already shared similar patterns of wood and wood fuel production. A few very small countries and territories (Andorra, Niue, Norfolk Island, and Saint Pierre and Miquelon) were excluded from clustering because their populations were close to zero and few predictor variables were available. We believe they should share the same general mechanisms as neighbouring areas, and so they were manually assigned to the cluster of the surrounding countries and territories.

As clustering was based on temporally correlated variables, a time series clustering approach was used. It was implemented with the tclust function in the package dtwclust version 5.5.6[49] for R Statistical Software[50]. The implementation was made using a hierarchical approach with dynamic time warping distance (DTW) and all types of hierarchical linkages. The optimal number of clusters was selected based on internal validity indices.

Clustering was applied in two stages. In the first stage, the number of clusters was allowed to vary from 3 to 10. In the second stage, clustering was applied within each cluster identified in the first stage, and the number of clusters was allowed to vary from 2 to 5. We used multinomial logistic regression to assess the influence of the 10 potential predictor variables on the partitioning.

In the first stage, three clusters were selected representing, generally, North America & Europe & North Africa & Central Asia, South America & Russia & Oceania, and Africa & South Asia. In the second

stage, two clusters were selected from each of the original three clusters for a total of six clusters, one of which contained only the Russian Federation. We eliminated it from further modelling because there are many years of wood fuel data available for the Russian Federation from international organizations and no similar countries from which to create new estimates. This left five clusters of countries and territories for modelling (Fig. 2, Supplementary Table 2). We note that the clustering algorithm did not assign "China, Hong Kong SAR", "China, Macao SAR", "China, Taiwan Province of", and "China, mainland" to the same cluster. Therefore, they were all manually assigned to the North America & Europe & China cluster where China, mainland was originally assigned by the algorithm.

We refer to the final clusters as "Central Africa & South Asia", "Southern Africa", "North America & Europe & China", "North Africa & Central Asia", and "South America & Oceania". However, we note that many more countries and territories belong to the clusters than are specifically included in the cluster name (Fig. 2, Supplementary Table 2). The most influential variables in cluster assignment were related to development (e.g., Percent Urban and Life Expectancy) and climate (e.g., Minimum Temperature and Rainfall).

### Regression and prediction

Our selection of predictor variables for regression aimed to represent our conceptual model of mechanistic drivers of wood fuel removals (Fig. 1) while also considering the international coverage of available data and the need to minimize inclusion of highly correlated variables, which could result in over-fitting and poor prediction performance in countries or territories with no wood fuel removal or wood charcoal production data. For example, access to electricity (Electricity) was not used for modelling as it had positive/negative Spearman correlation coefficients of 0.8–0.9 with GDP per capita, Biomass Use, and Life Expectancy, which were all used for modelling. In final modelling, we used a set of 12 predictor variables, described with data sources and definitions in Supplementary Table 1.

Our approach to imputation and smoothing of predictor variables is detailed in Supplementary Information, as well as creation of the per capita wood fuel demand and charcoal fraction response variables for regression. We used random forests as the methodological framework because they are flexible, deal relatively well with expected non-linear relationships and remaining correlation across predictor variables, and are relatively simple to implement. We fit one random forest model to each of the five country clusters (Supplementary Table 2). Random forests were fit using the ranger function in the R package ranger version 0.16.0[51], using 10,000 trees.

We selected values for two random forest tuning settings, the number of variables to possibly split at in each node (called "mtry" in ranger), and the minimal node size ("min.node.size") to split at, using two tenfold cross-validation procedures, each repeated ten times. The first procedure splits the data into ten roughly equal compartments by sampling random rows of the data, meaning data from a given country or territory can be spread across multiple test data sets. The second procedure splits the data into ten roughly equal compartments from the list of country and territory names for which at least one data point is available. The first procedure is therefore designed to represent prediction performance for countries and territories where some data may be available, while the second procedure is designed to represent prediction performance for countries and territories with no data. We selected a single value for each of the two tuning parameters (mtry = 4 and min.node.size = 5) to use for all random forest models, by considering the mean absolute error for wood fuel demand per capita and the charcoal fraction and then choosing a combination that we felt offered a defensible compromise in prediction performance between the two out-of-sample prediction procedures (generally, higher mtry values reduced mean absolute errors in the first procedure and increased mean absolute errors in the second procedure).

We used the final fitted random forests to predict per capita wood fuel and charcoal demand as a fraction of wood fuel demand for all years (1999–2019), countries and territories. We then converted these back to national estimates of wood fuel removals and charcoal production. Where imports were greater than the estimated demand plus exports, we set the estimated production to 0.

### Model composition

We assessed the influence of each predictor variable using ranger to compute permutation variable importance scores for each predictor variable (Fig. 6). The score for each variable is the decrease in prediction performance when values for that predictor variable are randomly shuffled. For informational purposes, we also computed the percentage of trees for which each variable appeared in the nth split, from the 1st split up to the 6th (Supplementary Figs. 2 and 3).

### Model limitations

The cost of up-to-date accurate estimates of wood removals for wood fuel and of wood charcoal production is exorbitant. Therefore, models will always be necessary for providing estimates where empirical data are insufficient or unavailable. We detail a credible and transparent modelling approach which, nevertheless, has limitations. As discussed in the future work section above, global disruptions, for example, are not included in real-time.

While we attempted to track all data sources and allow each to enter the model only once, it is possible that some values from household surveys are embedded in the model twice; once through the estimates of the proportion of the population using biomass or charcoal[10,52] and again through the literature search conducted for this study.

Uncertainty intervals, or similar, around final model predictions are not displayed as these would reflect only one component (regression uncertainty) of what is likely a tremendous level of uncertainty. There is also uncertainty, for example, in imputed values which was calculated and explored (Supplementary Information), but not tracked through the modelling process. Further uncertainty comes from the cut-offs of 10 m³ per capita wood fuel consumption and 0.1 tonnes per capita charcoal consumption, which were applied to data found through the literature and internet search. These cut-offs likely exclude some realistic values where there is a high level of undocumented trade, and likely allow quite a few unrealistic values below the cut-offs. The uncertainty in model estimates resulting from the choice of these cut-offs was not estimated. Our modelling approach also implies a dependence on trade data to move between production and demand, which adds uncertainty, particularly for those countries or territories where there is a high volume of trade and/or limited monitoring capacity.

Uncertainties inevitably accumulate along complex modelling paths similar to those of the garden of forking paths phenomena in hypothesis testing[53]. In this case, they accumulated not only from the imputation of missing data, data use thresholds, and trade data manipulations as described above but also from the selection of a modelling approach, the choice of input variables, and decisions related to clustering (unknown direction of bias). Additional uncertainties come from transformations, e.g., from stere to solid m³ (biases estimates slightly high), and definitions, e.g., wood fuel versus fuel wood (unknown direction of bias).

Likely the largest sources of uncertainty in our estimates are those related to measurement error of empirical surveys, including both measurement error at the household level (may not bias estimates) and in quantifying unregulated removals (likely biases estimates low), as well as estimation errors for values based on energy budgets (unknown direction of bias). Future work quantifying and communicating these uncertainties will add value both to the use of current estimates and to future modelling efforts.

We note that estimates for the Russian Federation were not revised because the clustering analysis indicated that the mechanisms driving wood fuel removals and wood charcoal may be too different from other countries to make pooling reasonable. Future estimates could be created through linear interpolation of official values.

An evolving model limitation concerns the future use by countries and territories of these modelled estimates to revise their official submissions to FAO. Where modelled estimates become official data or are used to adjust official estimates, they could then be fed into model updates, creating a loop in which it becomes impossible to untangle new information from iterated estimation procedures. We strongly encourage all users of these estimates to use them only where official data based on in-country modelling or in-country empirical data are not available, rather than as a source of information on which to base official data.

## Reporting summary
Further information on research design is available in the Nature Portfolio Reporting Summary linked to this article.

## Data availability
Source FAOSTAT data are publicly available at https://www.fao.org/faostat/en/#data/FO. Found data from the external data search (transformed and scaled for use in modelling) are available in Supplementary Data 3. Both the retained FAOSTAT and found data have also been deposited in Zenodo, along with smoothed covariate data and cluster membership labels, and are available for public download under a Creative Commons Attribution 4.0 International license[54].

## Code availability
R code to implement the random forest regression, cross-validation, prediction, and preparation of final results have been deposited on Zenodo and are available for public download under a Creative Commons Attribution 4.0 International license[54].

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

## Acknowledgements

The authors gratefully acknowledge the participation of Giovanna Jona Lasinio, Department of Statistical Sciences, Sapienza University of Rome; Zakia Adam, Arthur Contejean, Julia Guyon, Thierry Odou, and Zira John Quaghe, International Energy Agency (IEA); Heather Adair-Rohani, World Health Organization (WHO); Grace Bediako, Ghana Statistical Service, Arvydas Lebedys and Zuzhang Xia, Food and Agriculture Organization of the United Nations (FAO), Forestry Division; Adrian Whiteman, formerly International Renewable Energy Agency (IRENA); and Alexandra Schmidt, McGill University for committed participation in the expert working group to support data understanding, conceptual model development and statistical modelling. We also thank Matt Fonseca, formerly United Nations Economic Commission of Europe (UNECE), for expertise in forest product conversions; Marcella Canero, FAO Forestry Division, for identification of trade data to fill gaps; and Charlie Kirkwood, University of Exeter, for discussions on random forest methodology. Arturo Gianvenuti and Sven Walter, FAO Forestry Division contributed insight and support to the long-term success of the work. The authors gratefully acknowledge funding from the University of Glasgow's EPSRC Impact Acceleration Account [EP/X525716/1] awarded to Oliver Stoner.

## Author contributions

E.A.S. conceived of and led the project, working group, analysis, and manuscript preparation. H.P. and O.S. conducted the random forest modelling and assessment. O.S. combined and enhanced available code, ran and tested the final models and designed the final graphics. I.B. conducted the variable imputation and clustering analysis as well as supported the systematic data search. B.P. supported the systematic data search, data conversions, and development of a data management system. S.L.S. designed and initiated the systematic data search. E.P. initiated the idea of a conceptual model and prepared approaches for the clustering analyses. E.A.S., O.S., B.P., I.B., S.L.S., E.P., H.W., S.G., R.S., N.E., R.B., F.S. and L.R.S. participated in the expert working group, contributed expertise to guide the conceptual model and model interpretation, and contributed to manuscript preparation and review. The views and opinions expressed are those of the authors and do not necessarily reflect the official policy or position of the Italian National Institute of Statistics (Istat) or of the Food and Agriculture Organization of the United Nations (FAO).

## Competing interests

The authors declare no competing interests.
