## [Transparent Peer Review file · Nature Communications]

Global wood fuel production estimates and implications

Corresponding Author: Dr E. Steel

Version 0:

Reviewer comments:

Reviewer #1

(Remarks to the Author)

This paper modeled wood fuel and charcoal consumption for six global geographical clusters and used the estimated models to further estimate the country level annual wood removals and charcoal production. Overall, the manuscript is well written, is based on sound methodological framework, and demonstrates some novelties. The paper could be an important addition to forest products sector literature. However, before it can be accepted for publication, I see a need to correct one fundamental issue here. The issue is that the wood fuel and charcoal demand models that the authors' have estimated do not consider prices. A wood fuel demand model without its price as a predictor variable is unacceptable from the economics theory viewpoint.

The authors note that the expert group decided not to include price data because it would be scarce at a global level and not necessarily relevant at a national scale. However, data scarcity is not a valid reason to exclude a key predictor variable from the model. Also, price is always relevant regardless of the geographical scale. There are ways to derive a reasonable estimates of wood fuel and charcoal unit prices using FAOSTAT data. Models such as GFPM and FOROM, which the authors have cited in this paper estimate wood fuel prices for individual country using FAOSTAT data. Authors can use the similar approach to estimate country level unit prices of wood fuel and charcoal to include in their models and revise the results accordingly.

I have attached a few additional specific comments within the manuscript file.

Reviewer #2

(Remarks to the Author)

I am impressed when reading this manuscript, which presents a comprehensive study for an important and complex research topic. The amount of work is tremendous with many challenges. In general, the paper is well written with a proper article structure. However, I do have some concerns that need to be addressed (see below).

General comments:

1. The description for the section of the predictor variables is insufficient, though it is difficult when so many variables are considered. A possible way to simplify the description is to do it by major category of variables, as illustrated in Figure 1. And the variables in Figure 1 and Table 2 do not match completely. For example, the trade of wood fuels is not represented in Table 2, but illustrated in Figure 2. So, is trade considered or not? If not, why?
2. The study is done by country cluster, so the clustering of countries is important. However, I don't see sufficient description of the clustering, including the selection of the ten variables and the rationale of the variable selection.
3. Global trade of wood fuels, e.g., wood pellets from North America to Europe, can affect the relationships among wood fuel availability, demand, and consumption. In such a global study, the global trade of wood fuels needs to be considered in the modelling, and in Discussion section of "Implications for global energy and forestry policy".
4. In the developed countries, the implementation of climate mitigation policy can greatly affect wood fuel production and consumption. For example, the implementation of EU's RED has been a major driver for the greatly increased wood pellets consumption for energy production since 2009. Mitigation policies have resulted in elimination of coal-fire power generation

and increased use of wood bioenergy as one of the alternatives in some countries. Thus, such policies play an important role in determining the demand, production, and consumption. The authors need at least to discuss the limitation of not considering policy impacts.

5. It would be helpful for some readers if the authors provide the rationale for choosing machine learning, instead of more conventional modelling methods, to develop their models in this study.

6. While the full terms for most acronyms are given when used for the first time, even for those commonly used ones such as GDP, there are also some acronyms not specified, including LPG, CO₂, and some more specific terms used in this study (e.g., pp, percent trees, and many used in Figure 5 that are not defined/specified). Maybe you could mention that it is defined later in Table 2 when such a term, e.g., percent trees, is first used, or used in a figure/table (to make them stand-alone). By the way, GPD is defined several times (e.g., Line 150, 452).

Minor comments:

1. Line 92: add a comma between “industrial heating” and “electricity generation”.

2. Line 98-99: Using the term in the Paris Agreement, “National Determined Contributions”, is better than “national commitments” for the acronym “NDCs”?

3. Line 140: Delete “we” in “...resulting in we a new...”.

4. Line 142: Change “...from 1999-2019” to “from 1999 to 2019”.

5. Line 150: 1) Are the amount of people, land, and money” are country-specific here? If so, say it explicitly; and 2) should forest area is a more important variable than land area for wood fuel availability and removal?

6. Line 153-154, “Changes in forest area and cover might reflect wood fuel removals or charcoal production and were therefore also included as possible predictors”:

There might be two issues here: 1) Forest area and cover determine wood fuel availability and are more important than changes in forest area and cover. Thus should be included as possible predictors. 2) Changes in forest area and cover can affect wood fuel availability. But saying the changes reflect wood fuel removal or charcoal production might not be accurate. For example, main drivers for global deforestation include agricultural expansion, and urban and infrastructure development, which may not reflect wood fuel removal at all.

7. Line 163: Add “temperature” here: “Precipitation/temperature”.

8. Line 166 and 582-583: For clustering countries, which ten variables are selected from those listed in Table 2? And why?

9. Line 167: in “...identified from the conceptual model (Figure 2)”, should it be Figure 1?

10. Line 175-178: Specify why these countries displayed in white were not included in analysis?

11. Line 183-184: Add unit for the mean absolute errors, m³ per capita?

12. Line 222: What is “percentage trees”?

13. Line 236: Change “5.14pp” to “5.14 percentage points (pp)”.

14. Line 252: To make it clearer, add “with our estimates” before “potentially reflecting ...”.

15. Line 257-258: “... a lower estimate in total wood removals for producing fuel and a simultaneous increase in charcoal production”: should the wood removed for charcoal production be a part of the total wood removals for producing fuel?

16. Line 263: Change “other industrial roundwood production” to “the production of other industrial roundwood (OIR)”?

17. Line 281-282: ambiguous statement in “...wood volume is removed from forests and trees outside of forests...”. I guess you can make it clearer by changing to “...wood volume is removed from forests and from trees removed outside of forests...”.

18. Line 288: Add “in” here: “... the importance of wood energy for heating in some regions”.

19. Line 425: It is difficult to prove, I think, the listed assumptions are incorrect, since the factors and mechanisms driving wood fuel production are complex, and the two studies cited here might be just case studies compared to your study with a global scale. And I think it is a problem from oversimplification for the complex mechanisms that determine wood fuel production. Take the first assumption as an example, it makes sense to assume the total energy consumption in general increases with population; but the types of energy consumed and changes are determined by many other factors. So, it might be safer to say “inaccurate” instead of incorrect.

20. Line 465: there are two “production” in this sentence.

21. Line 725: I guess not everyone is familiar with Gini Index. So, it can be helpful to add a brief description of it.

Reviewer #3

(Remarks to the Author)

Thank you for the opportunity to review this article. It is a well-written, comprehensive study, with robust findings. I have only a few suggested comments and minor edits in the revised manuscript attached. The work will be important to improve our understanding of societal use of wood as a fuel, with implications also for greenhouse gas accounting.

Version 1:

Reviewer comments:

Reviewer #1

(Remarks to the Author)

Authors have addressed my comments.

Reviewer #2

(Remarks to the Author)

Dear authors,

Thank you very much for addressing my concerns, or otherwise enhancing the description. I am pleased with the revision of the manuscript.

Best regards,

Jiaxin

Reviewer #3

(Remarks to the Author)

I am satisfied by the answers from the authors. Congratulations on an important paper.

Summary of response to reviewers

We are grateful for the detailed and thoughtful feedback from all three reviewers. We have made changes in the manuscript in response to almost every single comment (point by point responses in blue below). The only major suggestion we did not incorporate was to include price estimates as predictors in the model. Our detailed explanation is below. In investigating whether there were any options to include price data in the model, we found opportunities to fine-tune the modeling process and this has substantially improved model performance. These model revisions have affected all tables and figures and so we synthesis these changes here for all reviewers.

Model revisions include:

- Revision of the data imputation procedure for missing predictor variables to better reflect data structure;
- Revision of modelling procedure to allow for multiple found data sources from the same year and country;
- Reduction of the number of predictor variables through elimination of two highly correlated predictors. Although random forest models manage correlation relatively well, through out-of-sample prediction experiments we found that model behavior could be improved by eliminating these variables. In this process, we also converted two potential predictors to be on a per capita basis;
- We re-tuned the random forest models by conducting an additional out-of-sample prediction experiment to assess performance for countries where some data are available and also for countries where no data are available;
- Where found data on wood fuel consumption were recorded as firewood (i.e., exclusive of charcoal), related found data on charcoal consumption were converted to wood volume and added to the firewood values, so that total wood removals for fuel is better captured;
- Addition of a screening criteria to eliminate data points that resulted in near zero per capita consumption for countries where more than 50% of the population is estimated to rely primarily on wood for cooking. Deleting implausibly low values for Pakistan and Bangladesh resulted in global estimates of wood fuel removals increasing by several hundred million cubic meters.

Overall, model predictive performance was improved and some unrealistic annual fluctuations within countries were eliminated.

All figures and tables describing model results have been updated; insertion of new tables and figures was not tracked in track changes to allow ease of viewing the document and reduce issues with Word managing the document. All text edits including table and figure legends were tracked in track changes.

Reviewer #1 (Remarks to the Author):

This paper modeled wood fuel and charcoal consumption for six global geographical clusters and used the estimated models to further estimate the country level annual wood removals and charcoal production. Overall, the manuscript is well written, is based on sound methodological

framework, and demonstrates some novelties. The paper could be an important addition to forest products sector literature. However, before it can be accepted for publication, I see a need to correct one fundamental issue here. The issue is that the wood fuel and charcoal demand models that the authors' have estimated do not consider prices. A wood fuel demand model without its price as a predictor variable is unacceptable from the economics theory viewpoint.

The authors note that the expert group decided not to include price data because it would be scarce at a global level and not necessarily relevant at a national scale. However, data scarcity is not a valid reason to exclude a key predictor variable from the model. Also, price is always relevant regardless of the geographical scale. There are ways to derive a reasonable estimates of wood fuel and charcoal unit prices using FAOSTAT data. Models such as GFPM and FOROM, which the authors have cited in this paper estimate wood fuel prices for individual country using FAOSTAT data. Authors can use the similar approach to estimate country level unit prices of wood fuel and charcoal to include in their models and revise the results accordingly.

We very much appreciate the importance of price where such data are available. In this case, we made a strong and thoughtful decision not to include price and we continue to believe this is the correct decision for the following reasons.

First and foremost, this is a model of wood fuel and charcoal production at national levels which includes as response data the strongest available data on FAOSTAT and which will likely be used by FAO to fill in missing FAOSTAT data. The available estimates of price from GFPM or FOROM would be based on estimates of wood fuel and charcoal production from FAOSTAT. Including price in this model would, therefore, effectively mean building a model using the response variable also as a predictor. We firmly believe such a model would be invalid from a statistical perspective and would create a circular relationship between modelling of production and modelling of price that increasingly magnifies biases and errors over time.

Second and relatedly, because we have the strongest FAOSTAT data and the available trade data in our model, the idea – the correlation structure underlying price – is already included. Adding price as modeled from data already in the model would effectively double the weight of this information without adding enough new information.

Finally, when we say that price data at a national scale would be invalid, we are describing the high spatial and temporal variance of prices across a country (e.g. across urban and rural areas and also across areas near and far from production etc.) and the challenge of effectively summarizing price at a national scale because such national summaries would require spatial and temporal weighting which are unavailable.

We have added two sentences in the paper to explain these ideas as concisely as possible.

I have attached a few additional specific comments within the manuscript file. [The following comments were captured from those embedded in the manuscript file].

Introduction

- List the full names of high-level global models - done

- Explain why the approach FAO was using for 20 years is no longer sufficient
 - o We added the following text “We bring information collected and statistical developments from the last 20 years to advance these wood fuel removal and wood charcoal production estimates.” We also note the model history section in the methods in we describe the necessarily constant cycle of revisions.

Results

- Population reads better than ‘amount of people’
 - o We had aimed to make a parallel between people, land, and money but we re-wrote using population and it is much clearer now. Thank you.
- Clarify whether the land here refers to total land area or forest land area. I think it refers to forest land, wooded land, and tree land
 - o It refers to the conceptual model and this was confusing because the last sentence of the paragraph switched topics inappropriately. We edited the sentence to make it clearer that the idea is a concept not a predictor variable and we removed the term predictor variable from the end of the paragraph. The exact model structure with predictor variables is detailed in the methods and the predictor variables are all identified, described and sourced in Table 2.
- Income perhaps is the better word here
 - o The entire sentence was edited as above, and the word was changed to economy.
- Price should be one important variable in any demand or supply model. A wood fuel demand model without its price as a predictor variable is unacceptable from the economics theory viewpoint.
 - o As described above, we disagree. This is a model in fact of demand based on a wide variety of factors that impact demand and including those factors used to estimate price. In fact, in all global models we are aware of, price is estimated from exactly the estimates we are creating.
- Figure 5 - Why the individual bar values are higher than the x-axis limit (1)? Shouldn't they be ≤ 1 ?
 - o Numbers do not reflect the individual bar values but rather the ranking. We added the following sentence to the legend: “Numbers to the right of each bar indicate ranking across variables within cluster. “
- Spell out percentage point - done
- Provide intuitive reasons for such outcomes. (charcoal fractions)
 - o As this is the results section, we do not provide an explanation or interpretation of model outcomes. The intuition and understanding of the results are provided in the discussion.

Discussion

- This is wood energy supply aspect. You are discussing wood energy demand here. Clarify how rainfall can influence wood energy demand.
 - o We added the phrase “Areas with relatively high levels of rainfall may also be more suitable for tree growth, which could be acting as a surrogate for overall wood fuel availability and therefore the possibility of using wood energy”

- Our previous understanding also suggests that price influence production or consumption? A demand model without its own price as a predictor variable is unacceptable from economic theory viewpoint.
 - o The above comments were confirming that the variables found in the model agree with our previous understanding of drivers of wood fuel consumption and production. Again, independent price data are not available and price was not included in the model.
- Your predictive model was for estimating consumption. So, describe factors affecting consumption.
 - o The purpose of the modeling exercise was to estimate wood fuel removals and charcoal production as produced by FAOSTAT. One of our innovations was to model this as demand per capita to best use new data from household surveys and to bring the effects of changes in population out of the model. As explained in the methods, the estimates of per capita demand/consumption were transformed into national estimates of production and this section discusses those new estimates.
- Suggest replacing it with 'necessitates' to avoid confusion. Done

Methods

- Specify whether you are referring to production modeling, consumption modelling or both?
 - o In this paragraph, we are discussing a history of modeling production as stated. In the following paragraph, we explain our new approach.
- Clarify how you estimated production through the demand for wood energy? Verify if the suggested clarifying text is correct. "In this paper, we estimate wood fuel production through the demand for wood energy by subtracting export quantity and adding import quantity to estimated consumption quantity (**Error! Reference source not found.**)"
 - o We kept the introductory line succinct and clear but, as suggested, added an explanation below. "The demand for wood energy is assumed to equal consumption and this is converted to production by adding exports and subtracting imports."
- Is found data wood fuel consumption or production data?
 - o We simplified and clarified as "We assumed that all found data represented wood fuel consumption when merging the values from FAOSTAT with found data"
- Spell out WHO
 - o We did not spell out WHO here as it was spelled out earlier in the paper.
- Data scarcity is not a valid reason to exclude a key predictor variable. There are ways to derive a reasonable estimate of unit price using FAOSTAT data. Models such as GFPM and FOROM, which you have cited in this paper, estimate wood fuel prices using FAOSTAT data.
 - o These models (GFPM and FOROM) use the estimates we are revising here as the key inputs; therefore, from a statistical perspective, this is an invalid approach. A full discussion of our decision not to include estimated priced data is above.
- Why would price data not be relevant at a national scale?
 - o Here we refer to empirical price data and have clarified as follows "And, finally, while unit prices could be derived in some areas using data from local studies, availability of such empirical estimates are scarce at a global level."
- These are four clusters, not three

- These are three clusters but the comma was confusing. We changed it to an “&” here and in Table 3 for all clusters to avoid confusion. This had already been done for the analysis cluster names.
- Why does it [dependence on trade data] add uncertainty to Europe particularly? And not much to other regions?
 - Because of the high volume of trade. We have clarified this in the text.

Tables and Figures

- Table 3 - Is this the second stage cluster? If so clarify, e.g. by providing in the parenthesis.
 - Thank you. Changed name to Countries and Territories in Analysis Cluster
- Change North America & Europe & China to North America, Europe & China
 - We did not make this change to keep labeling of clusters consistent and clear throughout.
- Change American to America - done

References

- Provide additional details to reference # 2, 9, 11, 14, 20, 30, 46, and 55. – done. Thank you very much.

Reviewer #2 (Remarks to the Author):

I am impressed when reading this manuscript, which presents a comprehensive study for an important and complex research topic. The amount of work is tremendous with many challenges. In general, the paper is well written with a proper article structure. However, I do have some concerns that need to be addressed (see below).

We thank the reviewer for acknowledging the work that went into this paper. It has been a long road and we are grateful for this positive response.

General comments:

1. The description for the section of the predictor variables is insufficient, though it is difficult when so many variables are considered. A possible way to simplify the description is to do it by major category of variables, as illustrated in Figure 1.

The variables, including data source and description are in Table 2 and sorted conceptually as suggested. We also kept variables from the same data source together. These are globally available data and fairly straightforward. The implications are considered in the discussion. We see that a clear statement however was never made and so we added this text “Our selection of predictor variables for regression aimed to represent our conceptual model of mechanistic drivers of wood fuel removals (Figure 1) while also considering the international coverage of available data and the need to minimize inclusion of highly correlated variables, which could result in over-fitting and poor prediction performance in countries with no wood fuel removal or wood charcoal production data. For example, access to electricity (Electricity) was not used for modelling as it had positive/negative spearman correlation coefficients of 0.8-0.9 with GDP per capita, Biomass Use, and Life Expectancy,

which were all used for modelling. In final modelling, we used a set of 12 predictor variables, described with data sources and definitions in Extended Data Table 2.” We thank the reviewer for expressing this need for clarity.

And the variables in Figure 1 and Table 2 do not match completely. For example, the trade of wood fuels is not represented in Table 2, but illustrated in Figure 2. So, is trade considered or not? If not, why?

Figure 1 is a conceptual model and Table 2 are the predictor variables which we used to build the statistical model. The conceptual model was used to select the best set of predictors and avoid overfitting or spurious correlation to the degree possible. This is described in the text and we now added the following to the legend for Figure 1. “The conceptual model for wood fuel production, describing the mechanistic drivers of wood fuel removals at the national level. These boxes do not represent the predictor variables in the model but rather the theoretical foundation for selection of predictor variables from globally available data. Final selection of predictor variables for clustering and for modelling is described in the Methods.” The text added to address the previous comment regarding selection of the predictor variables should also help to prevent confusion. Trade data were used to convert from demand/consumption to prediction as described in the Methods, e.g., “The demand for wood energy is assumed to equal consumption and this is converted to production by adding exports and subtracting imports.”

2. The study is done by country cluster, so the clustering of countries is important. However, I don't see sufficient description of the clustering, including the selection of the ten variables and the rationale of the variable selection.

This is an important omission, and the following text has been added to describe the clustering. “We used only 10 of the possible predictor variables because we first eliminated those that were highly correlated or that represented similar concepts to avoid overweighting the clustering by one idea or mechanism; we selected only one metric to describe rural versus urban population and we selected only Tree Area, which is the sum of wooded and forest area and highly related to country area. We also chose not to use metrics that were directly related to wood and wood fuel production for clustering, so that the clusters could describe countries that shared the same mechanisms driving wood fuel production and not countries which already shared similar patterns of wood and wood fuel production.”

3. Global trade of wood fuels, e.g., wood pellets from North America to Europe, can affect the relationships among wood fuel availability, demand, and consumption. In such a global study, the global trade of wood fuels needs to be considered in the modelling, and in Discussion section of “Implications for global energy and forestry policy”.

We note that wood pellets are a separate product in FAOSTAT and generally not produced from wood fuel removals but from sawdust and residuals of industrial roundwood removals, although this is evolving as noted below and in the paper. This presents a major conceptual gap in readers of the study and even in our expert working group – wood fuel modeling at the global level is quite different from generally Eurocentric-modeling of wood pellets. We have added the following text to the introduction: “modern wood-based fuels, including wood pellets and other agglomerates which are tracked separately in FAOSTAT but are

interlinked, ...” and the following text to the discussion to clarify “Increases in wood energy production may also be driven but other types of shifts in demand such as an increase in demand for modern wood fuels, innovations enabling new products or efficiencies in creating existing products such as wood pellets, and altered patterns of global trade.” We note that questions about exactly how wood pellets and wood fuel removals for wood pellets are reported reflect a complex and evolving issue requiring a lengthy explanation that is tangential to the findings reported here. The majority of cases where our estimates differ from values in FAOSTAT are from the developing world where the question has a lesser impact; therefore we have chosen not to emphasize the issue.

4. In the developed countries, the implementation of climate mitigation policy can greatly affect wood fuel production and consumption. For example, the implementation of EU’s RED has been a major driver for the greatly increased wood pellets consumption for energy production since 2009. Mitigation policies have resulted in elimination of coal-fire power generation and increased use of wood bioenergy as one of the alternatives in some countries. Thus, such policies play an important role in determining the demand, production, and consumption. The authors need at least to discuss the limitation of not considering policy impacts.

We believe that we have noted the importance. See here: Our new estimates agree with our existing understanding of wood removals for fuel and wood charcoal production across Europe; despite being slightly lower than current FAOSTAT values, our estimates continue to demonstrate an increasing trend. The trend is likely driven by low-carbon mandates leading to increased demand for pellets³⁹. Based on this increasing demand, there has also been a shift in the feedstock for producing wood pellets. Where wood pellets were originally produced almost entirely from industrial residues such sawdust, shavings, and particles, they may now be produced from an assortment of sources, including up to 50% low-quality roundwood⁴⁰.” Our paper, with limited word count, focuses on the revised estimates of past trends and implications of this new understanding. Most of the revised understanding is in the developing world. While there are many factors in Europe that have influenced wood energy, they are relatively well-understood and well-captured with existing values.

5. It would be helpful for some readers if the authors provide the rationale for choosing machine learning, instead of more conventional modelling methods, to develop their models in this study.

We have provided this explanation now slightly enhanced, “We used regression forests as the methodological framework because they are flexible, deal relatively well with expected non-linear relationships and remaining correlation across predictor variables, and are relatively simple to implement.”

6. While the full terms for most acronyms are given when used for the first time, even for those commonly used ones such as GDP, there are also some acronyms not specified, including LPG, CO₂, and some more specific terms used in this study (e.g., pp, percent trees, and many used in Figure 5 that are not defined/specified). Maybe you could mention that it is defined later in Table 2 when such a term, e.g., percent trees, is first used, or used

in a figure/table (to make them stand-alone). By the way, GPD is defined several times (e.g., Line 150, 452).

LPG defined, pp removed except in Supplement 2 where it is now defined, CO2 removed, GDP now defined once in text and at each use in a table or figure. We did not express everywhere that a variable was defined in Table 2 but clarified everywhere when we were referring to variable names.

Minor comments:

All suggestions were implemented. Thank you for the thorough check of these details. Where a response was necessary it was added in blue.

1. Line 92: add a comma between “industrial heating” and “electricity generation”.
2. Line 98-99: Using the term in the Paris Agreement, “National Determined Contributions”, is better than “national commitments” for the acronym “NDCs”?
3. Line 140: Delete “we” in “...resulting in we a new...”.
4. Line 142: Change “...from 1999-2019” to “from 1999 to 2019”.
5. Line 150: 1) Are the amount of people, land, and money” are country-specific here? If so, say it explicitly; Added “at national level” here and to the conceptual model legend and 2) should forest area is a more important variable than land area for wood fuel availability and removal? Land area was removed
6. Line 153-154, “Changes in forest area and cover might reflect wood fuel removals or charcoal production and were therefore also included as possible predictors”:
There might be two issues here: 1) Forest area and cover determine wood fuel availability and are more important than changes in forest area and cover. Thus should be included as possible predictors. 2) Changes in forest area and cover can affect wood fuel availability. But saying the changes reflect wood fuel removal or charcoal production might not be accurate. For example, main drivers for global deforestation include agricultural expansion, and urban and infrastructure development, which may not reflect wood fuel removal at all.
Changed to “might be correlated with”
7. Line 163: Add “temperature” here: “Precipitation/temperature”.
Temperature is already listed up with heating. Although it also could be a driver of availability, we chose not to list it twice.
8. Line 166 and 582-583: For clustering countries, which ten variables are selected from those listed in Table 2? And why?
All ten identified in Table 2 were used. These were the ten that seemed to describe the mechanisms driving energy demand and the fraction of that demand met with wood fuel. Detailed in the methods section
9. Line 167: in “...identified from the conceptual model (Figure 2)”, should it be Figure 1?
Corrected and figure numbers re-checked throughout.
10. Line 175-178: Specify why these countries displayed in white were not included in analysis?
There were a wide variety of reasons as described in the methods. Reference to the methods was added.
11. Line 183-184: Add unit for the mean absolute errors, m³ per capita?
Added for wood fuel (m³ per capita) and for charcoal (percentage points).
12. Line 222: What is “percentage trees”?

It reads as Percent Trees and it is a predictor variable as defined in Table 2. Variable names were capitalized throughout for clarity

13. Line 236: Change “5.14pp” to “5.14 percentage points (pp)”.

As this was the only use, removed acronym altogether

14. Line 252: To make it clearer, add “with our estimates” before “potentially reflecting ...”.

Clarified. Changed from FAOSTAT estimates to FAOSTAT values everywhere to clarify.

15. Line 257-258: “... a lower estimate in total wood removals for producing fuel and a simultaneous increase in charcoal production”: should the wood removed for charcoal production be a part of the total wood removals for producing fuel?

These are the way the data are reported and defined. There is a small inconsistency embedded in international reporting – the inconsistency between what is removed (wood fuel) and what is produced (charcoal).

16. Line 263: Change “other industrial roundwood production” to “the production of other industrial roundwood (OIR)”?

17. Line 281-282: ambiguous statement in “...wood volume is removed from forests and trees outside of forests...”. I guess you can make it clearer by changing to “...wood volume is removed from forests and from trees removed outside of forests...”.

Standard language but agree it is confusing – edited to “including trees outside of forests”

18. Line 288: Add “in” here: “... the importance of wood energy for heating in some regions”.

19. Line 425: It is difficult to prove, I think, the listed assumptions are incorrect, since the factors and mechanisms driving wood fuel production are complex, and the two studies cited here might be just case studies compared to your study with a global scale. And I think it is a problem from oversimplification for the complex mechanisms that determine wood fuel production. Take the first assumption as an example, it makes sense to assume the total energy consumption in general increases with population; but the types of energy consumed and changes are determined by many other factors. Changed to “more or less inaccurate” instead of incorrect.

20. Line 465: there are two “production” in this sentence - corrected

21. Line 725: I guess not everyone is familiar with Gini Index. So, it can be helpful to add a brief description of it. – Added

Reviewer #3 (Remarks to the Author):

Thank you for the opportunity to review this article. It is a well-written, comprehensive study, with robust findings. I have only a few suggested comments and minor edits in the revised manuscript attached. The work will be important to improve our understanding of societal use of wood as a fuel, with implications also for greenhouse gas accounting.

Thank you for the positive review of the value of our work.

From PDF – additional comments

Abstract

- Change “is” to “was”

Done

Introduction

- Need to be consistent with terminology - wood is inherently renewable, perhaps the authors meant "unsustainably derived wood fuels"?
Thank you. We went back to the original article and corrected to "with wood fuels harvested in a non-renewable manner."
- This sentence is a bit unclear - are the emissions due to forest degradation / deforestation, or are emissions from the use of the fuel / charcoal?
Forest harvest. We broke the sentence into two separate sentences to avoid future confusion.
- Important to highlight heat and power are only one example of use
By using the word "typically" we feel we have clarified this.
- Deleted "we"
done

Results

- While I fully agree with the idea of clustering, it is not immediately clear what the rationale was for clustering countries - more explanation is required. For example South America and Oceania?
Clarified that this was done using a statistical model and added an explanation of the logic "We assume that countries that are similar with respect to these 10 variables also share similar mechanisms driving energy demand and the fraction of that demand met with wood fuel. Creating models by cluster then enables pooling of information between similar countries and stronger predictions for countries with insufficient data." We note that there is no reason that countries would be clustered similarly except that they share similar suites of variables so, when breaking the globe into a few clusters, it is not surprising that some seemingly disparate areas and countries are found in the same cluster. We did not explain this idea because we feel it becomes evident.
- Any commentary on why newly revised estimates are lower
They are now higher – as we would have expected originally
- Any commentary on potential climate change impacts on future consumption trends
Not here in the results section and, in general, this is interesting but beyond the scope of our model. Added an idea about climate to the discussion lead-in from this comment – thank you.
- Technology for charcoal production varies significantly - is that a variable that could be included in the future? Also relatedly, market for biochar is predicted to increase globally, for both energy and non-energy needs. Can / should this be captured in future versions of the model? May be good to include some discussion on those points
We don't see how we could incorporate the variability in charcoal production technology at the global scale as there are no data to support it. Fascinating idea and maybe a small bit is captured with our indicators of, say, poverty. BioChar is a different product all together and outside of the scope of our analysis at this point. It is produced from all forms of biomass, not only wood. The idea would be important for forecasting future trends but we don't believe and we don't have the evidence to support a statement that increasing demand for biochar has influenced trends from 1999-2019.
- "electricity" – not sure what this means here

It is a variable and listed in the table. We edited the early part of the sentence to clarify. "minimum temperature and rainfall were the variables with the strongest influence on predictive accuracy, followed by electricity." [note that this is now changed based on revised model]

Discussion

- This reads more like part of a summary section than Discussion
Prefer to leave a short concise statement of the most important findings as the lead paragraph to the discussion as this is the most read section of the paper after the abstract. We go on to discuss all the findings. We did mention climate now in a slightly different way than above but very useful idea.
- Does this mean that countries NDCs will need to be adjusted accordingly under the Paris Agreement?
We cannot comment on what countries may need to do with their NDC commitments; we discuss here how revised estimates may impact calculation of indicators.
- Add reference for "human population has not grown as much as anticipated in some areas."
Clarified to what we intended which does not require a reference "Decreases compared to previous models which were highly driven by population forecasts may also reflect differences between forecasted and observed human population trends"
- Need to be careful with definitions here - "low quality roundwood" may in many cases be also considered residue in the absence of a market. The production of pellets may be the only reason the low-quality roundwood which may be harvested as part of an integrated forestry operation is actually removed from the forest
Clarified to "Where wood pellets were originally produced almost entirely from industrial residues such sawdust, shavings, and particles, they may now be produced from an assortment of sources, including up to 50% low-quality roundwood."
- Potential further disaggregation options could include use of charcoal for siderurgical applications, which is especially important in countries like Brazil
- Fascinating. We read up on this but decided not to mention this particular need among the biggest future needs and the many specific possibilities for future disaggregation.

Methods

- Check reference that did not convert correctly into pdf format
It was a table reference that was incorrect. It has been corrected. Thank you.